



# In-cloud scavenging scheme for aerosol modules

Eemeli Holopainen[1], Harri Kokkola[1], Anton Laakso[1], and Thomas Kühn[1,2]

[1]Atmospheric Research Centre of Eastern Finland, Finnish Meteorological Institute, P.O. Box 1627, 70211 Kuopio, Finland
[2]Aerosol Physics Research Group, University of Eastern Finland, P.O. Box 1627, 70211 Kuopio, Finland

**Correspondence:** Eemeli Holopainen (eemeli.holopainen@fmi.fi)

**Abstract.**

In this study we introduce an in-cloud wet deposition scheme for liquid and ice phase clouds for global aerosol-climate models which use a size-segregated aerosol description. For in-cloud nucleation scavenging, the scheme uses cloud droplet activation and ice nucleation rates obtained from the host model. For in-cloud impaction scavenging, we used a method where

the removal rate depends on the aerosol size and cloud droplet radii. The scheme was compared to a scheme that uses fixed scavenging coefficients. The comparison included vertical profiles and mass and number distributions of wet deposition fluxes of different aerosol compounds and for different latitude bands. Using the scheme presented here, mass concentrations for black carbon, organic carbon, sulfate, and the number concentration of particles with diameters larger than 100 nm are higher than using fixed scavenging coefficients, with the largest differences in the vertical profiles in the Arctic. On the other hand, the

number concentrations of small particles show a decrease, especially in the Arctic region. These results indicate that, compared to using fixed scavenging coefficients, nucleation scavenging is less efficient and impaction scavenging is increased in the scheme introduced here. Without further adjustments in the host model, our wet deposition scheme produced unrealistically high aerosol concentrations, especially at high altitudes. This also leads to a spuriously long lifetime of black carbon aerosol. To find a better setup for simulating aerosol vertical profiles and transport, sensitivity simulations were conducted where aerosol

emission distribution and hygroscopicity were altered. The simulated vertical profiles of aerosol in these sensitivity studies were evaluated against aircraft observations. The lifetimes of different aerosol compounds were also evaluated against the ensemble mean of models involved in the Aerosol Comparisons between Observations and Models (AEROCOM) project. The best comparison between the observations and the model was achieved with the new wet deposition scheme when black carbon was emitted internally mixed with soluble compounds instead of keeping it externally mixed. This also produced atmospheric

lifetimes for the other species which were comparable to the AEROCOM model means.

## 1   Introduction

The estimated radiation budget of the Earth has large uncertainties, and a majority of these uncertainties are related to the uncertainties in the direct and indirect effects of atmospheric aerosol (IPCC, 2014). Aerosol particles can affect the climate directly by scattering and absorbing radiation and indirectly through aerosol-cloud interactions (Haywood and Shine, 1997;

Twomey, 1991; Albrecht, 1989). Thus, in order to estimate the radiation budget of the Earth correctly, aerosols and their physical properties affecting radiation and cloud formation have to be modelled realistically.





Black carbon (BC) is one of the aerosol compounds which has an effect on the Earth's radiation budget via absorbing solar radiation, accelerating the melting of snow and ice, and influencing cloud formation and life-cycle (Bond et al., 2013). A large fraction of BC is emitted through incomplete combustion which is due to anthropogenic activities (Bond et al., 2013). Due to

its ability to darken snow and ice covers, BC has been found to be a major warming agent at high latitudes (AMAP, 2015). In addition, it has been proposed that the mitigation of BC is one of the possible means to slow Arctic warming (Stone et al., 2014).

Transport of aerosol particles to remote regions with no or only small amounts of emitted particles, affects the local aerosol size distribution and composition (Rasch et al., 2000; Croft et al., 2010). In these areas, e.g. the Arctic, simulated aerosol

and especially BC concentrations differ from those observed, as the transport to these regions is modelled poorly (Bourgeois and Bey, 2011; Sharma et al., 2013; Kristiansen et al., 2016). In addition, BC vertical profiles affect the uncertainty of its forcing emphasising the need to improve BC vertical profiles in global aerosol-climate models (Samset et al., 2013). The vertical distribution of aerosol compounds is found to be affected by emissions, hygroscopicity, deposition and microphysical processes, of which wet removal can be the cause of one of the major biases in the models (Kipling et al., 2016; Watson-Parris

et al., 2019). Thus, one possible cause for problems in modelling long-range and vertical transport of BC is how wet removal of aerosol compounds is modelled (Bourgeois and Bey, 2011; Croft et al., 2016). Wet deposition processes are modelled very differently among global aerosol-climate models and, therefore, more research is needed to better parameterise and constrain wet deposition in models (Croft et al., 2009, 2010, 2016; Textor et al., 2006; Kipling et al., 2016).

Wet removal of aerosol particles from the atmosphere is a process where these particles are scavenged by hydrometeors

and then carried to the surface by precipitation (Wang et al., 1978). There are two kinds of wet deposition processes: in-cloud and below-cloud scavenging (Slinn and Hales, 1971; Rasch et al., 2000; Zikova and Zdimal, 2016). In the process of in-cloud scavenging, aerosol species can enter the cloud droplets or ice crystals through a nucleation process, when they act as cloud condensation nuclei (CCN) or ice nuclei (IN). This process is called in-cloud nucleation scavenging (Ladino et al., 2011). In the process called in-cloud impaction scavenging, aerosol particles can be scavenged through collision with ice

crystals or cloud droplets (Chate et al., 2003). Aerosol compounds are then removed from the atmosphere when these cloud droplets or ice crystals grow to precipitation sizes (Pruppacher and Klett, 1997; Croft et al., 2010). Below-cloud scavenging is a process where rain droplets or snow crystals, which precipitate from the cloud, sweep aerosol particles below the cloud through collision (Chate et al., 2011). Observational studies have shown that below-cloud scavenging is strongly dependent on the rain droplet or snow crystal size distribution (Andronache, 2003; Andronache et al., 2006).

In recent years it has become evident that more detailed descriptions of wet deposition in global climate models is important (Korhonen et al., 2008; Garrett et al., 2010; Browse et al., 2012). In addition to transport, wet removal can affect the Arctic aerosol size distribution and its seasonal cycle (Korhonen et al., 2008; Croft et al., 2016). Even though the processes involved in wet removal are well known, it is still difficult to represent them well in global climate models (Eckhardt et al., 2015). In order to realistically describe the wet removal processes, a thorough knowledge on microphysics of condensation and precipitation,

as well as aerosol microphysics, is needed (Rasch et al., 2000).





Here, we describe a new in-cloud scheme for wet deposition using physical parameterisations for nucleation and impaction scavenging in liquid and ice clouds. We further tested the sensitivity of our new scheme to assumptions in aerosol emissions distribution and hygroscopicity. The structure of the paper is as follows. In Sect. 2 we present details on in-cloud nucleation and impaction scavenging in general and introduce our new in-cloud nucleation scavenging scheme for liquid and ice clouds.

In addition, we present details on the aerosol module SALSA and its components, which we used to test and evaluate our new scheme and its sensitivity. In Sect. 2 we present the modifications performed for SALSA to include in-cloud impaction scavenging. In the same section, we also present the aerosol-chemistry-climate model setup which is used for testing the scheme on a global scale. In Sect. 3 we present the evaluation of our new scheme against a fixed scavenging coefficient scheme in terms of vertical profiles and wet deposition fluxes of different aerosol compounds. In addition, in the same section, we

evaluate the vertical profiles of different aerosol compounds from the sensitivity simulations against those from ATom aircraft campaigns (Wofsy et al., 2018). We also compare the wet deposition fluxes, of different aerosol compounds, from different sensitivity simulations to each other. Finally, we compare the lifetimes from all of the simulations to mean from several models in the Aerosol Comparisons between Observations and Models (AEROCOM) project.

## 2 In-cloud wet deposition scheme

In this section we will describe the in-cloud nucleation and impaction scavenging, for both liquid and ice phase clouds. For both of these cloud phases, the removal of aerosol particles is expressed in terms of a scavenging coefficient. The rate of change in the concentration of compound $l$ in size class $i$, $C_i^l$, due to in-cloud nucleation and impaction scavenging, for both liquid and ice clouds, is of the form

$$\frac{\Delta C_i^l}{\Delta t} = C_i^l f^{\mathrm{cl}} \left( \frac{(F_{i,\mathrm{nuc,liq}} + F_{i,\mathrm{imp,liq}}) f^{\mathrm{liq}} Q^{\mathrm{liq}}}{C_{\mathrm{liq}}} + \frac{(F_{i,\mathrm{nuc,ice}} + F_{i,\mathrm{imp,ice}}) f^{\mathrm{ice}} Q^{\mathrm{ice}}}{C_{\mathrm{ice}}} \right), \qquad (1)$$

where $F_{i,\mathrm{nuc,liq}}$ and $F_{i,\mathrm{nuc,ice}}$ are the fractions of activated particles due to nucleation scavenging in liquid and ice clouds, respectively, and $F_{i,\mathrm{imp,liq}}$ and $F_{i,\mathrm{imp,ice}}$ are the scavenging coefficients due to impaction scavenging in liquid and ice clouds, respectively (Croft et al., 2010). Furthermore, $f^{\mathrm{cl}}$ is the cloud fraction, $f^{\mathrm{liq}}$ is the liquid fraction of the total cloud water, $Q^{\mathrm{liq}}$ is the sum of conversion rate of cloud liquid water to precipitation by autoconversion, accretion and aggregation processes, $C_{\mathrm{liq}}$ is the cloud liquid water content and $f^{\mathrm{ice}}$, $Q^{\mathrm{ice}}$ and $C_{\mathrm{ice}}$ are the equivalent variables for ice (Croft et al., 2010).

### 2.1 In-cloud scavenging scheme for liquid clouds

The in-cloud process of nucleation scavenging refers to activation and growth of aerosol particles into cloud droplets (Köhler, 1936). When water vapor reaches supersaturation, a fraction of the aerosol population is activated to cloud droplets. After these cloud droplets have grown to precipitation size, the particles can be removed from the cloud through precipitation (Wang et al., 1978). The ability of an aerosol particle to activate to a cloud droplet depends on its size, chemical composition and the

ambient supersaturation (Köhler, 1936).





In aerosol modules of global climate models, the aerosol size distribution can be approximated by, for example, a modal or sectional discretisation, which effectively separates the size distribution into different size classes (Stier et al., 2005; Kokkola et al., 2018a). In each size class the fraction of activated particles can be calculated as the portion of particles that exceed the critical diameter of activation in that size class (Köhler, 1936; Croft et al., 2010). However, many models describe the nucleation scavenging by assuming a constant scavenging coefficient for different aerosol size classes (Stier et al., 2005; de Bruine et al., 2018; Seland et al., 2008).

The new in-cloud nucleation scavenging scheme for liquid clouds introduced here, calculates the scavenging coefficients of aerosol based on the fraction of activated particles in each size class, i.e. $F_{i,\mathrm{nuc,liq}}$ in Eq. (1). Thus, using the scheme requires that the atmospheric model incorporates a cloud activation parameterisation that calculates size segregated cloud activation. Such parameterisations are e.g. Abdul-Razzak and Ghan (2002); Barahona and Nenes (2007).

In-cloud impaction scavenging, for liquid clouds, is a process where aerosol particles collide with existing cloud droplets and are thereby removed from the interstitial air of the cloud (Chate et al., 2003). This aerosol scavenging by cloud droplets is based on coagulation theory, which quantifies the rate of removal. This is further used to define the scavenging coefficients by impaction (Seinfeld and Pandis, 2006). Commonly, these scavenging coefficients, for the full aerosol particle distribution, can be calculated as

$$F_{i,\mathrm{imp,liq}}(d_{\mathrm{p}},t) = \int\limits_{0}^{\infty} K(d_{\mathrm{p}}, D_{\mathrm{liq}}) n(D_{\mathrm{liq}},t) dD_{\mathrm{liq}}, \tag{2}$$

where $d_{\mathrm{p}}$ is the diameter of the aerosol particle, $D_{\mathrm{liq}}$ is the cloud droplet diameter, $K(d_{\mathrm{p}}, D_{\mathrm{liq}})$ is the collection efficiency between aerosol particles and cloud droplets and $n(D_{\mathrm{liq}},t)$ is the cloud droplet number distribution (Seinfeld and Pandis, 2006).

## 2.2 In-cloud scavenging scheme for ice clouds

In-cloud nucleation scavenging in ice clouds refers to the formation and growth of ice particles (Seinfeld and Pandis, 2006). When ice particles are formed, they can quickly grow into precipitation sizes and be removed from the cloud (Korolev et al., 2011). The formation of ice particles in the atmosphere usually requires an ice nucleus (IN), but they can also be formed without IN, if the temperature is very low (Hobbs, 1993). Aerosol particles which can act as IN are usually insoluble (Marcolli et al., 2007). In addition, large particles are more efficient in acting as IN than small particles (Archuleta et al., 2005).

The nucleation rate, $J_T$, which is the total number of ice crystals formed in a unit volume of air per unit time, can be expressed as the sum of the nucleation rate in a unit volume of liquid solution, $J_V$, multiplied by the total collective volume of aerosol particles in a unit volume of air, $V_t$, and the nucleation rate on a unit surface area of liquid solution, $J_S$, multiplied by the total collective surface area of aerosol particles in a unit volume of air, $S_t$ (Tabazadeh et al., 2002). However, experimental studies and thermodynamic calculations for the ice-water-air system suggest that the total number of ice crystals formed is dominated





by surface-based processes, so that $J_S S_t \gg J_V V_t$ (Tabazadeh et al., 2002). With this assumption the total nucleation rate can be simplified to

$$J_T = \frac{\Delta\text{ICNC}}{\Delta t} = J_V V_t + J_S S_t \approx J_S S_t, \tag{3}$$

Global models usually give the total in-cloud ice nucleation rate, which is here segregated into size-resolved nucleation rates. Since we assume that the amount of nucleated ice particles depends only on the aerosol surface area, the scavenging coefficient in ice-containing clouds in size class $i$ is proportional to the ratio between nucleation rate in the size class and the total nucleation rate. Thus, we get for the scavenging coefficient, for the ice-containing clouds, in each size class

$$F_{i,\text{nuc,ice}} = \frac{S_i}{\sum_j S_j} \frac{\Delta\text{ICNC}}{n_i}, \tag{4}$$

where $S_i$ are the surface area concentration of size class $i$, $\Delta\text{ICNC}$ is the ice crystal number concentration obtained from the ice cloud activation scheme and $n_i$ the number concentration in size class $i$. The total surface area in each size class is derived using the associated number or mass median wet aerosol radius.

## 2.3 SALSA

To test our new in-cloud wet deposition scheme and its sensitivity, we used the Sectional Aerosol module for Large Scale Application (SALSA) in our model simulations. SALSA is a very versatile aerosol microphysics module which has been implemented in several models of very different spatial resolution (Kokkola et al., 2018a; Tonttila et al., 2017; Andersson et al., 2015; Kurppa et al., 2019). To describe the aerosol population, SALSA uses a hybrid bin sectional approach for calculating the evolution of the size distribution (Chen and Lamb, 1994; Kokkola et al., 2018a). In SALSA the aerosol population is divided into two subregions regarding their size. The first subregion is from 3 nm to 50 nm and the second is from 50 nm to 10 μm. These subregions are further divided into size sections defining the minimum and maximum diameter of the particles. In each size section the aerosol particles are assumed to be monodisperse, and chemistry and different microphysical processes are calculated for each size section separately. In addition, the second subregion is divided into externally mixed soluble and insoluble populations. A more detailed description of the newest SALSA version, SALSA2.0, is presented in Kokkola et al. (2018a).

Originally, SALSA uses fixed scavenging coefficients, $F_i$, for different size classes $i$, in its wet deposition calculations. These coefficients include all the processes for in-cloud and below cloud scavenging (Bergman et al., 2012). The fixed coefficients, for stratiform and convective clouds with different phases (liquid, mixed and ice) and solubilities, are adapted for SALSA from the calculations presented by Stier et al. (2005), and they are presented in detail in Bergman et al. (2012). Here we refine the entire scavenging scheme by calculating the scavenging coefficients online.

We used the Abdul-Razzak and Ghan (2002) cloud activation scheme to derive the fraction of activated particles in each size class for our in-cloud nucleation scavenging calculations. However, the original activation scheme considers only the soluble





material in particles and therefore neglects any possible insoluble material (Abdul-Razzak and Ghan, 2002). For computing the amount of cloud droplets formed, this is a good assumption, as usually most CCN-sized particles contain a large fraction of soluble material. However, when the insoluble fraction is large (>0.99), the assumption may lead to an underestimation of scavenged particles. This is because large insoluble particles with thin soluble coating (for instance mineral dust) are indirectly

assumed to be fairly small and may thus fail to activate into cloud droplets. Therefore, we modified the Abdul-Razzak and Ghan (2002) activation calculations to account for the insoluble core in particles. The calculations are otherwise the same, but the critical supersaturation for each size class is calculated using Eq. (17.38) in Seinfeld and Pandis (2006). The supersaturation calculations, used in the Abdul-Razzak and Ghan (2002) cloud activation, for particles containing an insoluble core are presented in appendix A. As input for the in-cloud nucleation scavenging coefficients in ice clouds we used the ice crystal

nucleation scheme described in Lohmann (2002).

   As the in-cloud nucleation scavenging was changed into a more functional method we also needed to alter the calculation of the in-cloud impaction scavenging. We calculate the in-cloud impaction scavenging in SALSA, for liquid clouds, using the same method as described in Croft et al. (2010). This method computes in-cloud impaction as a function of aerosol particle size ($r_p$), median aerosol particle radius ($r_{pg}$) and cloud droplet radii ($R_{liq}$). Using this same information from our monodis-

perse size classes for aerosol particles, we can assume that each size class is a log-normal mode and the in-cloud impaction scavenging coefficients, for liquid clouds, are then obtained as

$$F_{i,\text{imp,liq}} = \Lambda_\text{m}\left(r_\text{pg}\right)\Delta t, \tag{5}$$

   where $\Lambda_\text{m}\left(r_\text{pg}\right)$ is the mean mass scavenging coefficient, and it is defined as

$$\Lambda_\text{m}(r_\text{pg}) = \frac{\int_0^\infty \Lambda(r_\text{pg})r_\text{p}^3 n(r_\text{p})dr_\text{p}}{\int_0^\infty r_\text{p}^3 n(r_\text{p})dr_\text{p}}, \tag{6}$$

and

$$\Lambda(r_\text{pg}) = \int\limits_0^\infty \pi R_\text{liq}^2 U_t(R_\text{liq})E(R_\text{liq},r_\text{pg})n(R_\text{liq})dR_\text{liq}, \tag{7}$$

   which is called the scavenging coefficient in inverse time (Croft et al., 2010). In Eq. (6) and Eq. (7) $n(r_\text{p})$ is the aerosol number, $R_\text{liq}$ is the cloud droplet radius, $U_t(R_\text{liq})$ is the terminal velocity of cloud droplets, $E(R_\text{liq},r_\text{pg})$ is the collision efficiency between the aerosol particles and cloud droplets, and $n(R_\text{liq})$ is the cloud droplet number (Croft et al., 2010).

The in-cloud impaction scavenging, for ice clouds, is calculated following Croft et al. (2010), but as our model assumes that the ice crystals are monodisperse, there is no need to integrate over ice crystal number distribution (Croft et al., 2010). Thus, the in-cloud impaction scavenging coefficients are

$$F_{i,\text{imp,ice}} = \pi R_\text{ice}^2 U_t(R_\text{ice})E(R_\text{ice},r_\text{pg})\,\text{ICNC}\,\Delta t, \tag{8}$$





where $R_{\mathrm{ice}}$ is the radius of the ice crystal in its maximum extent, $U_t(R_{\mathrm{ice}})$ is the terminal velocity of the ice crystals and
$E(R_{\mathrm{ice}}, r_{\mathrm{pg}})$ is the collection efficiency of the collisions between aerosol particles and ice crystals (Croft et al., 2010).

## 2.4 ECHAM-HAMMOZ

For testing the effect of the new wet scavening scheme on global aerosol properties, we used the latest stable version of
ECHAM-HAMMOZ (ECHAM6.3-HAM2.3-MOZ1.0), a 3-dimensional aerosol-chemistry-climate model. ECHAM6.3 is a
general circulation model (GCM) and it solves the equations for divergence, temperature, surface pressure and vorticity (Stier
et al., 2005). These large-scale meteorological, prognostic, variables can be nudged towards data from operational weather
forecast models (Stier et al., 2005; Kokkola et al., 2018a).

ECHAM6.3 is coupled with Hamburg Aerosol Model (HAM), which calculates all of the aerosol properties within ECHAM-
HAMMOZ. These properties include emissions, deposition, radiation and microphysics (Stier et al., 2005; Tegen et al., 2019).
HAM has a comprehensive parametrisation for both modal and sectional microphysics representations of aerosol populations.
In addition to BC, the aerosol compounds included in this study are: organic carbon (OC), organic aerosol (OA) (here assumed
to be 1.4 times the modelled OC mass), sulfate ($SO_4$), mineral dust (DU) and sea salt (SS). ECHAM6.3 is further coupled
to the chemistry model MOZ (not used here) which contains a detailed stratospheric and tropospheric reactive chemistry
representation for 63 chemical species, including nitrogen oxides, tropospheric ozone and hydrocarbons (Schultz et al., 2018;
Horowitz et al., 2003).

## 195   2.5   Simulations

We used a total of 6 different simulations to investigate the performance of the new wet deposition scheme. The first two
simulations were done with default wet deposition scheme of SALSA (hereafter referred to as "old") and the wet deposition
scheme introduced in this study (hereafter referred to as "new"). As will be shown later, in the default model configuration the
new scheme resulted in spurious BC vertical profiles. To investigate the reasons for this, we carried out 4 additional sensitivity
simulations where we changed the assumptions of emission size distribution, as well as internal mixing and ageing of BC. An
overview over the different simulations, and their illustrative colors and line styles in the upcoming figures, are presented in
Table 1.

In the model simulations, the runs "baserun_new" and "baserun_old" are used to compare the new and old in-cloud scav-
enging schemes. The simulations "BC_small", "BC_large", "BC_soluble", and "insol2sol" were conducted to evaluate the
sensitivity of the new in-cloud scavenging scheme. These sensitivity studies were chosen based on the findings of Kipling et al.
(2016) who studied how model processes affect the simulated aerosol vertical profiles. Their study indicated that the processes
which have the strongest effect on aerosol vertical profiles in the HadGEM model are emission distribution, hygroscopicity,
deposition and microphysical processes (Kipling et al., 2016).

In the first two sensitivity runs, we altered the BC emission distribution for SALSA. This was done so that all of the BC
emissions were directed to either size class of small or large insoluble particles, respectively. In the default configuration the
BC emission size distributions are log-normal mass fraction distributions following AEROCOM emission recommendations





**Table 1.** Overview of the simulations used in this study.

| Setup | Description | Illustration |
|---|---|---|
| baserun_old | Old ECHAM-SALSA in-cloud scavenging scheme with fixed scavenging coefficients. | — · — |
| baserun_new | New in-cloud nucleation scavenging using Abdul-Razzak and Ghan (2002) for liquid clouds and Lohmann (2002) for ice clouds. In-cloud impaction for liquid and ice clouds according to Croft et al. (2010) | —— |
| BC_small | All BC emissions directed to small insoluble size class. | ······· |
| BC_large | All BC emissions directed to large insoluble size class. | — · · |
| BC_soluble | All BC emissions directed to soluble population with the same mass distribution as for baseruns. | — · — · |
| insol2sol | Simulating ageing of insoluble particles by moving them to soluble aerosol population after they activate at 0.5 % supersaturation. | — — · |

(Stier et al., 2005; Dentener et al., 2006), which are remapped to the SALSA size classes. The mode radii ($r_m$) and standard deviations $\sigma$ for the original BC emission size distributions are $r_m = 0.015$ µm and $\sigma = 1.8$, for fossil fuel emissions, and $r_m = 0.04$ µm and $\sigma = 1.8$, for wild-fire emissions (Dentener et al., 2006). In the BC_small simulation, we directed all BC

emissions to an insoluble size class where particle diameter spans from 50 nm to 96.7 nm. In the BC_large simulation, we directed all BC emissions to an insoluble size class where particle diameter spans from 0.7 µm to 1.7 µm.

To study the sensitivity of the wet deposition scheme to BC hygroscopicity, we conducted a simulation where all BC emissions were directed to soluble size classes. The size distribution for the emissions was the same as for the baserun simulations when they are directed to the insoluble classes. This simulation is referred to as BC_soluble in the model simulations. In the

fourth sensitivity study, called insol2sol, insoluble particles are transferred to parallel size classes of soluble particles. This allows for separation of fresh and aged particles and is a method to simulate aerosol ageing used also in other global aerosol models (e.g. Stier et al., 2005). The criterion for transfer is that particles activate at a supersaturation of 0.5 %. A schematic of the aerosol emission distribution for the different simulations is presented in Fig. 1.

## 2.6 Experimental setup

The simulations were performed with ECHAM-HAMMOZ for the year 2010, with the SALSA aerosol module, using 3-hourly data output, after a six-month spin-up. The emissions were obtained from the ACCMIP (Emissions for Atmospheric Chemistry and Climate Model Intercomparison Project) emission inventories which are interpolated, for the period 2000-2100 by using Representative Concentration Pathway 4.5 (RCP4.5) (Lamarque et al., 2010; van Vuuren et al., 2011). The model vorticity, divergence and surface pressure were nudged towards meteorological observations of ECMWF (European Centre for Medium-

Range Weather Forecasts) (Simmons et al., 1989), and the sea surface temperature (SST) and sea ice cover (SIC) were also



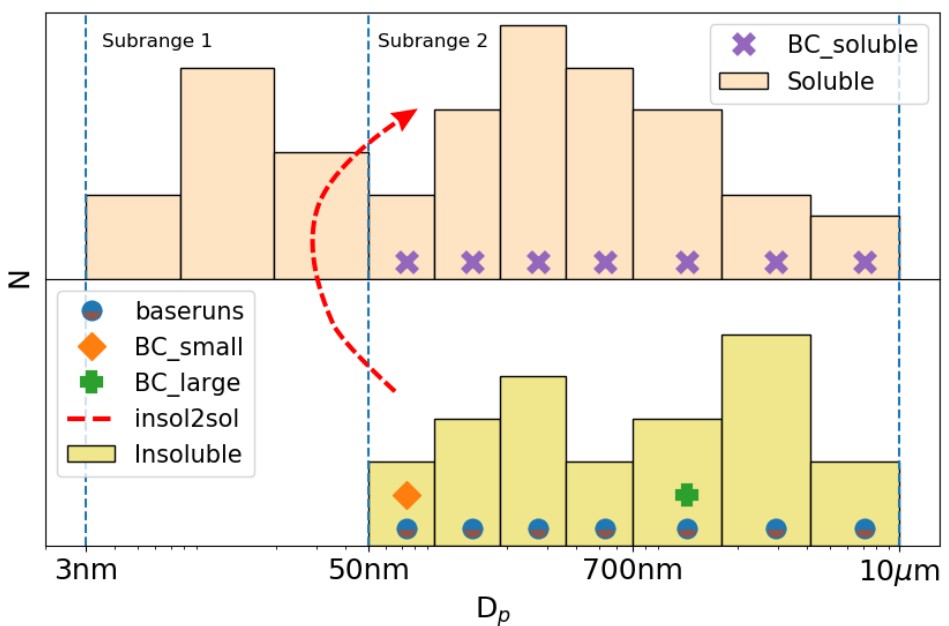

**Figure 1.** Schematic representation of the distribution of aerosols in different simulation runs.

prescribed. SST and SIC were obtained from monthly mean climatologies from AMIP (Atmospheric Model Intercomparison Project). The analysis is made between the old and the new wet deposition scheme using SALSA. In addition, the sensitivity of the new scheme to emission sizes, aging, and hygroscopicity of BC-containing aerosol, is tested using ECHAM-HAMMOZ with SALSA.

## 2.7  ATom aircraft measurements

To see how the new scheme and the sensitivity studies reproduces the vertical properties of different aerosol compounds, we compared the model simulations against aircraft measurements. The aircraft data was obtained from all NASA's Atmospheric Tomography (ATom) missions (1, 2, 3, and 4), and the dataset was merged data from all instruments which measure atmospheric chemistry, trace gases, and aerosols (Wofsy et al., 2018).

To get the best representative comparison between the ATom aircraft measurements and model data, the model data was sampled to the same time and locations of the aircraft measurements. For the collocation of model vertical profiles with observations, we used the Community Intercomparison Suite (CIS) tool (Watson-Parris et al., 2016).

BC concentrations were measured with Single-Particle Soot Photometer (NOAA) (SP2) and OA and $SO_4$ concentrations with CU Aircraft High-Resolution Time-of-Flight Aerosol Mass Spectrometer (HR-AMS) (Wofsy et al., 2018). Number concentration of particles with diameter larger than 100 nm, $N_{100}$, and total number concentration, $N_{tot}$ were combined from the data measured with a nucleation-mode aerosol size spectrometer (NMASS), an ultra-high-sensitivity aerosol size spectrometer (UHSAS) and a laser aerosol spectrometer (LAS) (Brock et al., 2019; Wofsy et al., 2018).





## 3 Results

### 3.1 Differences between simulated values of old and new wet deposition schemes

First, we compared how aerosol properties differ when using the old and the new wet deposition schemes. In order to assess, how the two schemes affect aerosol transport and vertical profiles, we compared the modelled aerosol vertical profiles over the tropics (0-30° N), the mid-latitudes (30-60° N) and the Arctic (60-90° N). Here we focused on $SO_4$, OC (or OA), and BC as they are readily available from the ATom aircraft campaign measurements.

Figure 2 shows the vertical profile of BC, OC and $SO_4$ mass concentration simulated with the old and the new in-cloud wet deposition schemes. The different rows show different latitude bands, as horizontally averaged annual means. The figure illustrates that all three of the compounds show similar differences in the vertical profiles in all three latitude bands, between the two runs. The concentrations for each compound are higher for the new scheme compared to the old scheme for almost the entire vertical domain. The differences between the different wet deposition schemes are greatest at higher altitudes starting from approx. 900 hPa upwards. In the tropics, these differences in the profiles are lowest with a maximum relative difference of approx. 200 % for BC and OC and slightly exceeding 150 % for $SO_4$. These maxima occur at approx. 200 hPa altitude. In the mid-latitudes, the differences are slightly higher than at the tropics and the maximum difference in the values are at ∼300 hPa altitude. The new method shows ∼350 % higher concentrations at maximum for BC and $SO_4$ and ∼400 % for OC. The Arctic shows the greatest differences in the compound profiles. The difference is largest at ∼500 hPa altitude where the concentrations in the new scheme outweigh the concentrations in the old scheme by ∼600 % for BC, 650 % for OC and 800 % for $SO_4$. As emissions of these aerosol particles in the Arctic are low, most aerosol is transported into the Arctic from emission regions outside the Arctic. It is thus evident that the wet removal of these aerosol particles is reduced in the new scheme, which allows for the particles to be transported to higher altitudes and longer distances. In addition, we found that the model accumulates BC at the higher altitudes in simulations spanning several years (not shown), which can be considered spurious behaviour.

Figure 3 shows the vertical profile of the number concentration of particles with diameters larger than 100 nm, $N_{100}$, and the total number concentration, $N_{tot}$. The $N_{100}$ profiles show similar differences between the old and the new scheme as for the concentration profiles of different compounds in Fig. 2. In addition, the relative increase in the concentrations in the new wet deposition scheme is similar. This can be explained by less efficient nucleation scavenging in the new scheme which reduces the wet removal of large particles and thus increases the number concentration of large particles. Particles larger than 100 nm act as condensation sink for $H_2SO_4$ and thus an increase in $N_{100}$ leads to reduced new particle formation (NPF) and thus to decreased number concentrations of small particles. This can be seen in the $N_{tot}$ profiles, which show a decrease in the new scheme. This difference is most pronounced in the Arctic, where the relative difference between the new and old schemes in the $N_{tot}$ concentration reaches its maximum of ∼90 % at ∼400 hPa. In addition to large particles acting as condensation sink to gases, impaction scavenging is faster in the new scheme which in turn increases the removal of small aerosol particles and thus reduces concentrations even more. These effects become evident when looking at size-resolved wet deposition fluxes.



**Figure 2.** Vertical profiles of BC (left column), OC (center column) and $SO_4$ (right column), simulated with old and new in-cloud wet deposition schemes at different latitude bands. Note the different units for the different compounds.

The annual and global average size distribution of the wet deposition flux of the old and new in-cloud scavenging schemes are presented in Fig. 4. The wet deposition size distributions confirm what has been observed in the vertical aerosol profiles.

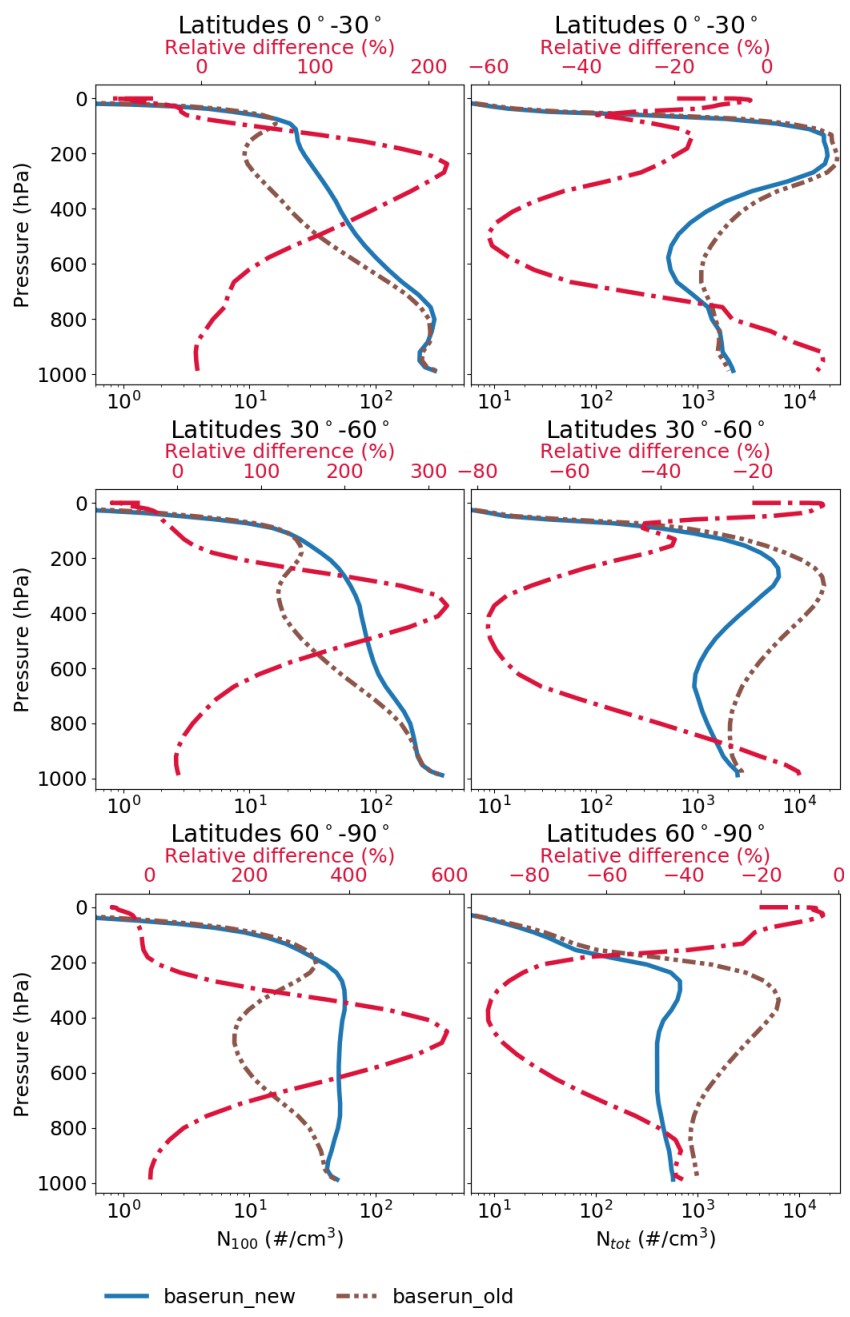

**Figure 3.** Vertical profiles of the $N_{100}$ (left column) and $N_{\text{tot}}$ (right column) concentrations, simulated with old and new in-cloud wet deposition schemes at different latitude regions.

There is a modest change in the mass fluxes between the old and the new schemes. As in steady state the total emissions of a compound must match its total removal, these differences mostly stem from changes in the interplay between dry and



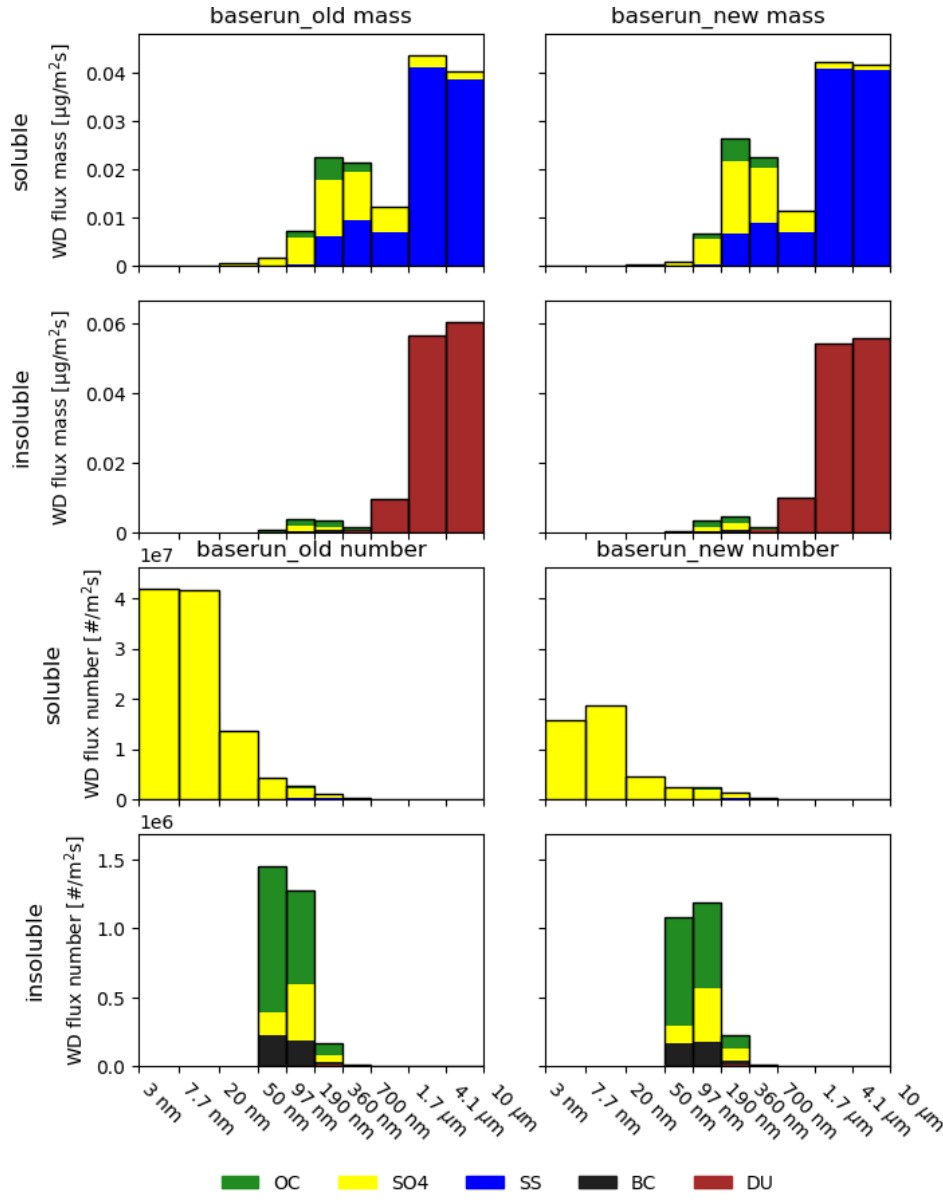

**Figure 4.** Wet deposition flux size distributions of different aerosol compounds simulated with old (left column) and new (right column) in-cloud wet deposition schemes. The top 4 figures show the wet deposition flux for the mass distribution and the lower 4 for the number distribution. Different rows show values for the different solubility types.

wet deposition processes. However, the number flux in the smallest size classes of the soluble population is halved, affecting mainly the removal of $SO_4$ in the smallest size classes. In addition, a small shift towards larger removed accumulation-sized particles can be observed (less particles are removed below 100 nm and more are removed above 100 nm). For the larger





size classes the decrease is more moderate. These results can be explained by increased concentrations of medium-sized and large particles in the new scheme which act as condensation sink for $SO_4$. This leads to less small particles as they are mainly

formed through NPF from gaseous $H_2SO_4$. This effect can also be seen in Fig. 4 as a slight increase in removed sulfate mass in the accumulation-sized particles of both the soluble and insoluble aerosol populations. As a consequence of the atmospheric concentration of small particles, the wet deposition flux for the smallest size classes is reduced in the new scheme compared to the old.

The lifetime of different aerosol compounds was calculated by dividing the annual mean global mass burden of each com-

pound by the annual mean emissions of the same compound (Lund et al., 2018). The global mean lifetime for BC was 9.23 days for the old scheme and 14.62 days for the new scheme. However, experimental studies from different aircraft campaigns indicate that the BC lifetime should be less than 5.5 days (Lund et al., 2018). This is a very interesting result: the more physical wet deposition scheme produces more spurious atmospheric lifetimes for BC. Consequently, also the ability of ECHAM-HAMMOZ to reliably simulate aerosol vertical profiles and long range transport of aerosol is decreased when using the more

physical scheme with the default model setup. This may be due to the fact that a more physical treatment of the wet deposition processes makes the model more sensitive to influences outside of the parameterisation. We therefore performed further sensitivity simulations and compared their results to observational data.

### 3.2    Sensitivity simulations

As reported in the previous section, ECHAM-HAMMOZ with the new, more physical scheme, in its default setup, produced

spurious BC vertical profiles. With the sensitivity simulations we aimed to explore different possibilities to improve the BC vertical profiles and long-range transport in the model. In order to increase nucleation scavenging of BC, we considered three different possibilities to make BC-containing particles more susceptible to cloud droplet activation. One way to achieve this is to emit BC into larger particles, which require less aging to be activated at a given supersaturation. This was tested in simulation BC_large. Another way is to mix BC with soluble compounds in order to enhance hygroscopicity of BC-containing particles

and thus their cloud activation susceptibility. This can be done in two ways, either by emitting BC directly to soluble size classes (simulation BC_soluble) or by emitting BC to insoluble size classes and transferring particles to soluble classes after aging (simulation insol2sol). A third way is to emit BC into smaller size classes in order to facilitate transfer of BC into larger, more easily activated particles by coagulation (simulation BC_small).

Figure 5 shows vertical profiles of BC, OA and $SO_4$ simulated with the new wet deposition scheme for the different sensitiv-

ity simulations, together with the average values from ATom aircraft measurements. The grey shaded area shows the standard deviation for the aircraft measurements. For BC, the sensitivity simulations BC_large, BC_soluble and insol2sol show a better match with observed vertical profiles than the other simulations in every latitude band. These simulations fall between the standard deviation limits of the ATom aircraft simulations almost everywhere, with exception of the tropics, where they underestimate the concentrations starting from ∼600 hPa downwards. In addition, in the tropics, BC_soluble and insol2sol

represent the BC concentrations slightly better than BC_large between 500 hPa and 300 hPa. BC_small and baserun_new overestimate the BC concentrations at all latitudes, except in the tropics at lower altitudes starting from ∼700 hPa downwards,



**Figure 5.** Vertical profiles of BC (left column), OA (center column) and $SO_4$ (right column), modelled with different sensitivity studies, compared to the ATom aircraft measurements at different latitude bands. Right of every subplot is the number of observations measured by the device, at each vertical level, from the ATom aircraft measurement campaigns. Note the different units for the different compounds.

where they represent the BC concentrations slightly better than the other sensitivity simulations. As we saw in the previous chapter, the reduced efficiency in the wet deposition increases BC concentrations at higher altitudes which causes baserun_new




to overestimate the BC concentrations. This is because the default emission sizes of BC particles are not very susceptible to
cloud activation. In addition, although BC_small aimed at increasing BC wet removal by emitting BC to small particle sizes
and thus enhancing their collection by coagulation to large particles, it is apparent that coagulation is not very efficient in doing
so.

Compared to baserun_new, most of the sensitivity studies show better agreement of the modelled BC profiles with the
measurements. However, it needs to also be checked if these solutions work as well for OA and $SO_4$. At all latitude regions
OA concentrations from all of the sensitivity simulations show similar results as the measurements, except for the insol2sol
simulation, which underestimates OA concentrations in the mid-latitudes as well as at higher altitudes in the tropics and the
Arctic. In the tropics the insol2sol simulation underestimates OA concentrations starting from approx. 700 hPa upwards. At
most insol2sol underestimates the measurements at the highest altitudes, in all of the latitude bands, where the concentrations
are over 1 order of magnitude less than the measurements. As the aging of aerosol particles in insol2sol is simulated by moving
all insoluble particles that can activate to cloud droplets at 0.5% supersaturation, almost all OA that is originally emitted to
insoluble size classes is moved to soluble size classes. Thus, this enhances the activation and consequently the wet deposition
of OA. Faster wet removal reduces the amount of OA transported to higher altitudes and thus reduces the OA concentrations.
OA concentrations from all other simulations fall between the standard deviation limits of the ATom aircraft measurements
everywhere, with only a slight overestimation between approx. 900 hPa and 800 hPa in the tropics.

For $SO_4$, all of the sensitivity simulations show similar trends as the measurements, but overestimates concentrations almost
everywhere. The effect that insol2sol has on OA concentrations is also visible in the $SO_4$ profiles, but here the effect is
much weaker. In the tropics, insol2sol shows better agreement with the measurements from 700 hPa upwards, than the other
simulations, with only a slight overestimation. Between approx. 900 hPa and 700 hPa, all of the simulations overestimate
the measurements. This may be due to simplified sulfate chemistry in the model as $SO_4$ is mainly formed through chemical
transformation. In the mid-latitudes, all simulations overestimate the $SO_4$ concentrations, with the exception of insol2sol
which reproduces the $SO_4$ profile slightly better than the other simulations from approx. 600 hPa upwards. However, near
the surface, all simulations overestimate the $SO_4$ concentrations by approximately half an order of magnitude. In the Arctic,
all of the simulations have similar $SO_4$ profiles with a slight overestimation between approx. 700 hPa and 300 hPa altitude.
However, at the highest altitudes all of the simulations underestimate the $SO_4$ concentrations. The different sensitivity tests
do not alter the $SO_4$ concentrations much compared to baserun_new, because most of it condenses onto soluble particles.
In addition, the new particles formed through nucleation are added to the soluble aerosol population. Thus, the $SO_4$ vertical
profiles are similar in all of the simulations, with an exception of insol2sol where some of the $SO_4$, which repartitions from the
insoluble to the soluble population, is activated more efficiently.

Figure 6 shows the vertical profiles of $N_{100}$ and $N_{tot}$, simulated with the sensitivity studies, together with ATom aircraft
measurements. From the figure we can see that $N_{100}$ profiles between different simulations are similar in the mid-latitudes
and the Arctic. In these latitude bands, the simulations slightly underestimate the $N_{100}$ concentrations when compared to the
measurements, but the trend is similar throughout the entire vertical column. However, insol2sol underestimates the $N_{100}$
profiles slightly more in the mid-latitudes and the Arctic. In the tropics, the simulations show a good correlation with the







**Figure 6.** Vertical profiles of the $N_{\mathrm{tot}}$ and $N_{100}$ concentrations, modelled with different sensitivity studies, compared to the ATom aircraft measurements at different latitude regions.

measurements, except for the surface concentrations, which are underestimated by a factor of about 2.5 compared to the
360 measurements. In addition, in the tropics, insol2sol underestimates the $N_{100}$ more than the other simulations from 800 hPa





upwards. This is also due to more efficient activation compared to baserun_new for medium-sized particles which reduces the transport to higher altitudes.

The $N_{\text{tot}}$ profiles are, all in all, fairly similar for all of the sensitivity simulations, with only modest differences. In the tropics the trend of the profiles varies between simulations and measurements. The sensitivity simulations tend to overestimate the $N_{\text{tot}}$ concentrations at the surface and at the highest altitudes by over 50 percent, but underestimate them at approx. 400-700 hPa. In the mid-latitudes, all of the simulations represent $N_{\text{tot}}$ concentrations fairly well when compared to the measurements. However, in the Arctic, all of the simulations underestimate the $N_{\text{tot}}$ profiles. At higher altitudes, starting from approx. 600 hPa upwards, insol2sol underestimates $N_{\text{tot}}$ least, showing quite a good agreement with the measurements.

One of the reasons for the differences in the $N_{\text{tot}}$ and $N_{100}$ surface concentrations may be due to a misrepresentation of the emitted particle size distribution. In ECHAM-HAMMOZ the same aerosol emission size distribution per compound and emission sector is assumed throughout the whole world, which is not very realistic for every aerosol particle source (Paasonen et al., 2016). At higher altitudes, the aerosol microphysical processes correct the aerosol size distribution towards more realistic profiles.

To investigate the effects of the different sensitivity studies further, we computed the size and mass distribution of the wet deposition flux (Fig. 7). The mass fluxes in the soluble population do not change much between baserun_new and the different sensitivity studies, except for the insol2sol simulation which allows for sufficiently hygroscopic particles of the insoluble population to be repartitioned to the soluble population. This leads to an increase in DU mass in the soluble population and a decrease in the insoluble population. In addition to more efficient wet removal of DU due to this process, this also increases dry deposition and sedimentation (not shown) of DU in insol2sol. For the mass fluxes in the insoluble population, BC_large and BC_soluble show an increase in the largest size class for DU. This effect is due to more efficient removal of BC-containing particles, which allows for more $SO_4$ to condense on larger, DU-containing particles, which enhances the activation of these particles.

The number fluxes for different sensitivity simulations in soluble population show most change in the two smallest size classes, which increases by a factor of about 1.3 in the insol2sol simulation and about 1.1 for BC_large and BC_soluble when compared to baserun_new. These differences stem from changes in medium-sized and large particle concentrations, which act as condensation sink for $SO_4$ and thereby regulate the amount of $SO_4$ available for new particle formation. In addition, there is a slight increase of OC in the insol2sol simulated number distribution, which is being transferred from the insoluble population. Otherwise, there is no notable change in other compounds as the $SO_4$ dominates the number distribution in the soluble population. The relative BC mass contribution to the wet deposition number flux of the insoluble aerosol population very well reflects the assumptions made in the different sensitivity studies. While for BC_large and BC_soluble the BC mass fraction in the medium-sized insoluble particles disappears, in BC_small the BC fraction in the 50 to 100 nm insoluble particles is about 3 times larger than in baserun_new. This shows that coagulation is not effective in moving BC from these small insoluble particles to large soluble particles. In insol2sol, most of the BC is moved to the insoluble aerosol population before removal, which can be seen in a strong decrease in removed insoluble aerosol number for that simulation.



**Figure 7.** Wet deposition flux size distributions of different aerosol compounds simulated with different sensitivity simulations. Each column represents a different sensitivity study and each row the solubility type. The top 2 rows show the mass size distribution of the wet deposition flux and bottom 2 rows the number size distribution.

In addition to the evaluation of the simulated vertical aerosol profiles, we used the modelled atmospheric lifetimes of all aerosol compounds as indicator of the model skill in the different simulations. Here we estimated the atmospheric lifetime of a compound as the yearly and global mean mass burden of the compound divided by its total yearly mean emission. The com-



piled mean lifetimes for the different simulations and compounds as well as the mean lifetimes from several AEROCOM models (CAM5-ATRAS, EC-Earth, TM5, ECHAM-HAM, ECHAM-SALSA, ECMWF-IFS, EMEP, GEOS, GFDL-AM4, GISS-
OMA, INCA, NorESM2, OsloCTM3 and SPRINTARS) are presented in Table 2 (Gliß et al., 2020).

**Table 2.** Lifetimes of compounds from different simulations and mean from different AEROCOM models.

|  | baserun_old | baserun_new | BC_small | BC_large | insol2sol | BC_soluble | AEROCOM |
|---|---|---|---|---|---|---|---|
| $\tau_{BC}$ (d) | 9.23 | 14.62 | 16.49 | 5.78 | 5.04 | 4.98 | 5.8 |
| $\tau_{DU}$ (d) | 4.07 | 5.36 | 5.69 | 5.00 | 1.06 | 4.86 | 4.5 |
| $\tau_{SO_4}$ (d) | 4.02 | 6.10 | 6.37 | 5.73 | 4.69 | 5.67 | 4.7 |
| $\tau_{OC}$ (d) | 6.38 | 9.44 | 9.52 | 9.03 | 4.90 | 8.90 | 6.1 |
| $\tau_{SS}$ (d) | 1.59 | 1.57 | 1.57 | 1.56 | 1.55 | 1.56 | 0.82 |

With the assumption that the AEROCOM mean atmospheric lifetimes are the current best guess, we can use Table 2 to select a simulation that best reproduces these mean lifetimes and therefore could be considered as the best solution to address the overestimated BC lifetimes in baserun_new. While baserun_old, baserun_new and BC_small overestimate the BC lifetime by factors of 1.6, 2.5 and 2.8, respectively, BC_large, insol2sol and BC_soluble all produce BC lifetimes within one day of

the AEROCOM mean. In addition, the BC lifetimes should be less than 5.5 days according to Lund et al. (2018). However, the different sensitivity studies also affect the atmospheric lifetimes of the other species, and some of them considerably. For instance, the lifetime of DU in insol2sol is almost 4.5 times shorter than the AEROCOM mean, while both BC_large and BC_soluble overestimate this mean only slightly by half a day. On the other hand, the atmospheric lifetime of OC in insol2sol is closest to the AEROCOM mean compared to all other simulations using the new wet deposition scheme. However, in this

setup of ECHAM-HAMMOZ all OC is emitted as primary particles, while in reality a large fraction of the organic aerosol is formed as secondary organic aerosol (SOA) in the atmosphere. Modelling the processes leading to SOA formation more realistically would most likely affect the modelled OC lifetimes quite substantially. Therefore we do not use the modelled OC lifetimes as skill indicator for the sensitivity studies here. The atmospheric lifetime of $SO_4$ in insol2sol is also closest to the AEROCOM mean, but also BC_large and BC_soluble model the $SO_4$ lifetime fairly well. For SS, the atmospheric lifetime

does not change when changing the wet removal algorithm or during any of the sensitivity tests as SS is only emitted to the soluble population. The lifetimes for all simulations are more than 0.7 days higher than the AEROCOM mean (about a factor of 2). This has already been discussed by Kokkola et al. (2018a) and Tegen et al. (2019).

## 4    Conclusions

We developed a new in-cloud nucleation wet deposition scheme for liquid and ice clouds. For liquid clouds, the scavenging
coefficients are calculated using the size-segregated fraction of activated particles from a cloud activation scheme. For ice



clouds, are calculated the scavenging coefficients based on the surface area concentration of each size class (see Tabazadeh et al., 2002).

We used the SALSA microphysics scheme coupled with the ECHAM-HAMMOZ global aerosol-chemistry-climate model to evaluate and test our new in-cloud wet deposition scheme. In its original setup, SALSA used fixed scavenging coefficients

for modelling wet deposition. Here, we used the Abdul-Razzak and Ghan (2002) cloud activation scheme for the calculations of size dependent nucleation scavenging coefficients in liquid clouds. For ice clouds, we used the scheme of Lohmann (2002) for providing the ice nucleation rates for the nucleation scavenging scheme (see Tabazadeh et al., 2002). The in-cloud impaction scavenging for SALSA was adapted from the method for modal scheme by Croft et al. (2010).

Compared to using fixed scavenging coefficients, the new scheme showed an increase in BC, OA, and $SO_4$ vertical profiles

almost throughout the entire vertical domain for all latitude bands. In the Arctic region this increase was most pronounced, with a maximum increase of up to 800 %. The differences in vertical profiles had similar functional shapes in all latitude bands and for all three compounds. The increase was mainly due to a decrease in the nucleation scavenging of aerosol particles in the new scheme, which increased aerosol transport into the upper atmosphere and subsequently to the Arctic region. The new scheme also showed a significant increase in large particle concentrations which was similar in shape to the change in

aerosol compound mass. However, the small particle concentrations decreased everywhere, with a maximum decrease of 90 % in the Arctic. This implies that new particle formation was reduced in the new scheme due the increased concentration of large particles, which increased the condensation sink for $SO_4$. In addition, impaction scavenging in the new scheme was faster which increased the removal rate of small particles even more.

An evaluation of the new wet deposition scheme against ATom aircraft measurements showed that, using the default setup

of the host model, the new scheme overestimated BC mass concentrations, especially at higher altitudes. Additional sensitivity simulations showed that the model skill of reproducing measured vertical BC mass concentration profiles could be improved a lot by directing the BC emissions to larger or to more soluble size classes, or by transferring BC-containing particles to soluble size classes after aging. These sensitivity studies also produced BC atmospheric lifetimes which were closest to the AEROCOM model mean Gliß et al. (2020). Emitting BC to smaller size classes, on the other hand, overestimated the aerosol

mass concentrations and BC atmospheric lifetime even more. However, changing the distribution of BC in the sensitivity simulations also affected the mass concentrations of other aerosol compounds. For instance, transferring insoluble particles to soluble size classes after aging led to an underestimation of the the observed OA concentrations at higher altitudes, while in the other simulations OA concentrations fell between the standard deviation limits of ATom measurements almost everywhere. The modelled atmospheric lifetime of OA, on the other hand, compared best to the AEROCOM mean when transferring aged

insoluble particles to soluble size classes. However, as in this study secondary processes of OA formation were neglected, we did not use OA as an indicator for the skill of our wet deposition scheme. For $SO_4$, the insoluble-to-soluble transfer reproduced the observed concentrations slightly better at higher altitudes in the tropics. Nevertheless, all simulations showed similar results for $SO_4$ concentrations, with only a slight overestimation when compared to the aircraft observations. In addition, $SO_4$ atmospheric lifetimes did not vary much across the different sensitivity studies. All of the sensitivity studies reproduced

aerosol number concentration profiles fairly well. However, the insoluble-to-soluble transfer considerably underestimated the





concentrations of activation-sized particles at the highest altitudes in the tropics, which was strongly tied to the underestimation of OC at these altitudes. Furthermore, the atmospheric lifetime of atmospheric mineral dust (DU) was strongly underestimated in the simulation using insoluble-to-soluble transfer of aged particles. The atmospheric lifetimes of seasalt (SS) did not change between the different sensitivity studies. All in all, while reasonable BC vertical profiles and atmospheric lifetimes could be

achieved with the new wet deposition scheme in three of the sensitivity studies, namely emitting BC to more hygroscopic or to larger particles or transferring insoluble, BC-containing particles, to soluble size classes, only the first option is really suitable. Emitting BC to large particles is quite unrealistic, because the emission size of BC-containing particles is fairly well established (Tissari et al., 2008; Krecl et al., 2017; Corbin et al., 2018; Zhang et al., 2019) and insoluble-to-soluble transfer, on the other hand, lead to too small atmospheric lifetimes of DU.

To conclude, even though the new in-cloud wet deposition scheme is more physically sound than using fixed scavenging coefficients, it failed to reproduces global aerosol fields adequately in the default setup of the host model. In particular, the BC atmospheric lifetime was almost 3 times as large as what observations indicate (Lund et al., 2018). Based on the results of our sensitivity simulations, model produces the best vertical profiles and aerosol lifetimes with the new scheme if BC is mixed with more soluble compounds at emission time.

*Code availability.* The ECHAM6-HAMMOZ model is made available to the scientific community under the HAMMOZ Software Licence Agreement, which defines the conditions under which the model can be used. The licence can be downloaded from https://redmine.hammoz. ethz.ch/attachments/291/License_ECHAM-HAMMOZ_June2012.pdf (last access: 29 June 2012, HAMMOZ consortium, 2012).

The stand-alone zero-dimensional version of SALSA2.0 is distributed under the Apache-2.0 licence and the code is available at https: //github.com/UCLALES-SALSA/SALSA-standalone/releases/tag/2.0 (last access: 23 May 2018, Kokkola et al., 2018b)

*Data availability.* The model data can be reproduced using the model revision r5511 from the repository https://redmine.hammoz.ethz.ch/ projects/hammoz/repository/changes/echam6-hammoz/branches/fmi/fmi_trunk (last access: 8 March 2019, HAMMOZ consortium, 2019). The settings for the simulations are given in the same folder, in folder "gmd-2020-220". Alternatively, the data and codes for figures can be obtained directly from authors or from https://etsin.fairdata.fi/dataset/f3cb5807-66fe-4a0d-a20a-ac208d3aab5a (last access: 29 June 2020, Holopainen et al., 2020). All other input files are ECHAM-HAMMOZ standard and are available from the HAMMOZ repository (see

https://redmine.hammoz.ethz.ch/projects/hammoz, HAMMOZ consortium, 2019).

ATom aircraft data can be obtained through the Oak Ridge National Laboratory (ORNL) Distributed Active Archive Center (DAAC) https://daac.ornl.gov/cgi-bin/dsviewer.pl?ds_id=1581 (last access: 25 November 2019, Wofsy et al., 2018)

## Appendix A: Calculations for particles containing an insoluble core

The calculations for the particle containing an insoluble core are based on the technical report by Kokkola et al. (2008) where
the critical supersaturation is obtained as

 

$$\frac{S_c}{A} = \frac{2\left(\frac{b}{3}\right)^2 + \left(\frac{b}{3}\right)\left[\left(\frac{\gamma_+}{2}\right)^{1/3} + \left(\frac{\gamma_-}{2}\right)^{1/3}\right] + \left[\left(\frac{\gamma_+}{2}\right)^{2/3} + \left(\frac{\gamma_-}{2}\right)^{2/3}\right]}{9\left(\frac{b}{3}\right)^3 + 6\left(\frac{b}{3}\right)^2\left[\left(\frac{\gamma_+}{2}\right)^{1/3} + \left(\frac{\gamma_-}{2}\right)^{1/3}\right] + 3\left(\frac{b}{3}\right)\left[\left(\frac{\gamma_+}{2}\right)^{2/3} + \left(\frac{\gamma_-}{2}\right)^{2/3}\right] + d} \tag{A1}$$

where

$$\gamma_\pm = \left[2\left(\frac{b}{3}\right)^3 + d\right] \pm \sqrt{4\left(\frac{b}{3}\right)^3 d + d^2} \tag{A2}$$

and

$$b = \sqrt{\frac{3B}{A}} \tag{A3}$$

$$d = D_{p,0}^3. \tag{A4}$$

In Eq. (A3) $A$ and $B$ are obtained from Seinfeld and Pandis (2006). $A$ describes the increase in water vapour pressure due to the curvature of the particle surface and is denoted as

$$A = \frac{4M_w\sigma_w}{R\rho_w T}, \tag{A5}$$

and $B$ is called the solute effect term and is denoted as

$$B = \frac{6n_s M_w}{\pi\rho_w}. \tag{A6}$$

Using this new expression for the critical supersaturation, the effective critical supersaturation, maximum supersaturation, and the number fraction of activated particles for each size size class can be calculated using Eq. (8), (9) and (12-15) from the Abdul-Razzak and Ghan (2002).

*Author contributions.* EH, TK and HK designed the outline of the paper. EH wrote the majority of the paper. EH performed all the climate simulations. EH, TK and HK developed the new wet deposition scheme. TK and HK provided the calculations for particles containing an insoluble core. EH and AL modified the emission distributions for the sensitivity simulations. EH, TK, HK and AL performed the data analysis for the climate simulations and EH produced the figures. All the authors contributed to the writing of the paper.

*Competing interests.* The authors declare that they have no conflict of interest.





*Acknowledgements.* ECHAM-HAMMOZ is developed by a consortium composed of ETH Zürich, Max Planck Institut für Meteorologie, Forschungszentrum Jülich, University of Oxford, the Finnish Meteorological Institute and the Leibniz Institute for Tropospheric Research, and managed by the Center for Climate Systems Modeling (C2SM) at ETH Zürich. We thank NASA / ORNL DAAC for Atmospheric Tomography Mission (ATom) aircraft data.





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
