# Peer review of "In-cloud scavenging scheme for sectional aerosol modules -Implementation in the framework of SALSA2.0 global aerosol module"

_Geoscientific Model Development, 2020_

## Short Comment (SC1) · 3 Aug 2020

Dear authors,

in my role as Executive editor of GMD, I would like to bring to your attention our Editorial version 1.2:

https://www.geosci-model-dev.net/12/2215/2019/

This highlights some requirements of papers published in GMD, which is also available on the GMD website in the 'Manuscript Types' section: http://www.geoscientific-model-development.net/submission/manuscript_types.html

In particular, please note that for your paper, the following requirement has not been

met in the Discussions paper:

- "The main paper must give the model name and version number (or other unique identifier) in the title."

- Code must be published on a persistent public archive with a unique identifier for the exact model version described in the paper or uploaded to the supplement, unless this is impossible for reasons beyond the control of authors. All papers must include a section, at the end of the paper, entitled "Code availability". Here, either instructions for obtaining the code, or the reasons why the code is not available should be clearly stated. It is preferred for the code to be uploaded as a supplement or to be made available at a data repository with an associated DOI (digital object identifier) for the exact model version described in the paper. Alternatively, for established models, there may be an existing means of accessing the code through a particular system. In this case, there must exist a means of permanently accessing the precise model version described in the paper. In some cases, authors may prefer to put models on their own website, or to act as a point of contact for obtaining the code. Given the impermanence of websites and email addresses, this is not encouraged, and authors should consider improving the availability with a more permanent arrangement. Making code available through personal websites or via email contact to the authors is not sufficient. After the paper is accepted the model archive should be updated to include a link to the GMD paper.

Thus add the model name and version number SALSA 2.0 in the title of your article.

We very much appreciate that, while HAMMOZ is license restricted, a stand-alone version of SALSA 2.0 is made available. However, please note, that GMD is demanding authors to provide a persistent access to the exact version of the source code used for the model version presented in the paper. As explained in https://www.geoscientific-model-development.net/about/manuscript_types.html the preferred reference to this

release is through the use of a DOI which then can be cited in the paper. For projects in GitHub a DOI for a released code version can easily be created using Zenodo, see https://guides.github.com/activities/citable-code/ for details.

Finally note, that according to our new Editorial (v1.2) all data and analysis / plotting scripts should be made available.

Yours, Astrid Kerkweg

---

## Referee Comment (RC1) · Anonymous Referee #1 · 13 Aug 2020

In this manuscript, a revised size-segregated in-cloud aerosol wet removal scheme is implemented in the Sectional Model for Large Scale Application (SALSA). The revised wet removal scheme determines the fraction of aerosol that is contained in cloud hydrometeors based on cloud droplet activation and ice nucleation rates, and also includes size-dependent in-cloud impaction scavenging. This scheme is compared to another scheme that uses fixed scavenging coefficients. The authors also examine sensitivity studies with varying assumptions about the size of black carbon emissions and hygroscopicity. Model output for the various simulations is compared with aircraft observations from the ATom campaigns.

The manuscript addresses parameterizations that are of key importance but are notoriously challenging for global models that simulate aerosol concentrations. The pre-

sented results are scientifically interesting and contribute towards development of wet removal parameterizations for aerosol modules. The manuscript should be suitable for publication in GMD provided that the following concerns can be satisfactorily addressed. Certain aspects of the model description and discussion lack clarity as noted in the specific comments below. Please consider revisions to improve the clarity of the presentation with careful attention to details. As well, to put the study in context of previous work, please consider including discussion about how these results compare to other studies that have introduced similar wet removal schemes, while identifying the aspects of this scheme that are novel. A clearer presentation of main recommendations for the global modelling community could also be of benefit to the manuscript.

Specific Comments

1) An identification of the model used in the study would be of help to readers of the abstract.

2) Line 10-11: Please clarify what sizes are meant by 'small particles'. As well, the number of these particles could be influenced by changes in the rate of new particle formation. As a result, it is not clear that this decrease in number concentration indicates that impaction scavenging is increased relative to the fixed coefficient scheme.

3) Lines 15-16: Why was the simulation baserun_old excluded from the comparisons with observations? Please consider including this simulation in comparisons with the observations.

4) Line 61: 'new in-cloud scheme' - Please consider clearly identifying the main aspects of the scheme that are new relative to previous studies. The word 'new' is used 5 times in this paragraph and repeatedly throughout the manuscript. It could be helpful to the readers to provide information that assists with understanding the developments made here relative to earlier work.

5) Eq. 1 – Are certain of these variables in-cloud versus grid-box mean?

6) Line 107: Is the aerosol diameter wet or dry for this calculation?

7) Line 113: What aerosols are ice nuclei in the model?

8) Line 133-135: Is there a model version number? Please clarify what you mean by 'its sensitivity'.

9) Line 149: 'refine the entire scavenging scheme'. Is below-cloud scavenging also modified? Are both convective and stratiform wet removal modified? Please provide clarification about the wet removal treatment for the stratiform versus convective clouds. Are there differences between these two? How is the cloud fraction parameterized for each for the purposes of wet removal? Are there differences in the assumed updrafts for cloud droplet activation for stratiform and convective clouds?

10) Line 154-155: What size is meant by 'large particles'? What size is meant by 'fairly small'? Did you conduct any test simulations for dust without the modified activation scheme but with the revised wet removal?

11) Eq. 5-7: Please clarify if this is wet aerosol radius.

12) Line 165: 'assume each size class is a lognormal mode' – for consistency is this same assumption also made for the nucleation scavenging? In that case, are separate scavenging coefficients calculated for mass versus number?

13) Eq. 7 and Eq. 8: Are there specific references for the collision efficiencies, terminal velocity and ice crystal radius used here?

14) Section 2.4: Please clarify if the SALSA module is coupled to ECHAM6.3-HAM2.3-MOZ1.0 for all simulations.

15) Section 2.3-2.5: Please consider adding a description about the treatment of aerosol aging in the baserun_old and baserun_new. Is there any exchange between the soluble and insoluble classes in the baserun simulations? Also consider adding brief discussion about the treatment for OA emissions, sulfate emissions and chem-

istry. As well, do the simulated particles grow by aqueous phase sulfate production? What is the treatment for removal of gas-phase particle precursors? Consider mentioning here that the model does not include secondary organic aerosol.

16) What is the treatment of below-cloud wet removal in these simulations? In the subsequent discussions, consider addressing how these parameterizations impact your conclusions. Likewise, what is the treatment of dry deposition and how does that affect your conclusions?

17) Please consider referring to Fig. 1 at the start of Section 2.5 to help the reader to better follow the details presented.

18) Table 1: The color and line style for baserun_old and BC_soluble are very similar. Please consider revising.

19) Line 213-214: Does the model also include biofuel emissions?

20) Line 232-234: Are the baserun_old and baserun_new simulations not coupled to ECHAM-HAMMOZ but the sensitivity simulations are coupled and why? The text did not appear clear on the related description. Are the cloud microphysics parameterizations that are relevant for the wet removal different between the baserun simulations and the sensitivity simulations?

21) Section 2.7: What are the size ranges for the SP2 and HR-AMS? Do you extract mass concentrations from the model with consideration to similar size ranges?

22) Line 245: What is the size range for the total number concentration?

23) Line 268: 'model accumulated BC' – why does this occur, over how many years, would the model eventually reach a steady state?

24) Line 279: 'impaction scavenging is faster' – how can this be determined? If impaction scavenging is implicit in the prescribed coefficients scheme, then impaction rates cannot be directly compared between the two schemes - and as well other processes such as new particle formation can affect the number concentration. A similar question arises regarding the statement at line 273 since nucleation scavenging rates cannot be directly compared if both nucleation and impaction are implicitly represented by the fixed coefficients.

25) Figure 2: Please consider using the same scale for the relative differences for all panels.

26) Figure 3: What is the size range for Ntot?

27) Figure 4: Do these plots include both stratiform and convective wet deposition?

28) Line 294: Are there observation-based lifetimes available from previous studies for OA and sulfate, in addition to the lifetimes for BC from Lund et al., 2018?

29) Line 305: Do you mean the global mean BC 'lifetimes' are spurious? The previous section did not show vertical profiles.

30) Figure 5: Please consider if it would be instructive to include baserun_old in this figure.

31) Line 335: How sensitive are the results to the assumed supersaturation?

32) Line 344: What is meant by 'simplified sulfate chemistry'?

33) Line 370: 'same aerosol size distribution' – please consider if this information should be in the methods – is this assumption used for all simulations?

34) Lines 392: Some of this discussion was confusing – '3 times larger than in baserun_new' – where is baserun_new shown in Fig. 7?

35) Line 419: Please consider presenting what are the new developments made with this wet deposition scheme relative to previous studies. As well, consider putting the study in context of previous work by presenting how the main findings of this study compare to previous similar model developments.

[Figure]

36) Line 425: Perhaps the following could be clarified in the methods – for all simulations does SALSA run with an on-line coupling to a certain version of ECHAM-HAM-MOZ model and are the outputs used for the wet removal from the cloud droplet activation scheme and ice nucleation scheme from the same ECHAM-HAM-MOZ?

37) Line 428: What are the main adaptations needed for this wet removal scheme relative to the Croft et al. (2010) scheme?

38) Line 437: Please clarify how you know that '..impaction scavenging in the new scheme was faster'. If impaction was implicit in the prescribed coefficients scheme and new particle formation also influences particle number concentrations as noted in the text – how can this statement be justified?

39) Line 460-462: The implementation of insoluble to soluble transfer is dismissed as being unsuitable. However, this aging process is commonly included with various parameterizations in global models. Are the authors able to clarify why the particular parameterization used in this study was chosen? Are there certain aspects of the parameterization that could be improved with future work to enable a representation of aging from the insoluble to soluble classes? Why did the chosen parameterization for aging perform so poorly for dust in these simulations? What are the emitted dust sizes?

40) Line 466: 'failed to reproduce global aerosol fields adequately...'. This statement is quite general. Please clarify. Does this statement refer to all aerosols – or it is specifically for BC? For example, at line 413, OC is excluded as a skill indicator.

41) Please consider including the main recommendations for future model development based on the findings of this study.

Technical corrections

1) Line 5: 'aerosol size' - please clarify if this is wet or dry aerosol radius

2) Line 33: 'no or small amount' – consider removal of 'no'

3) Line 95-96: Check order of citations

4) Line 48: Please check this citation as Ladino et al., 2011 appears to focus on impaction as opposed to nucleation scavenging.

5) Line 125: 'amount of nucleated ice particles' – do you mean number nucleated?

6) Figure 1: 'N' on vertical axis is not defined

7) Line 259 'are lowest' – do you mean relative to other latitudes?

8) Line 283: 'modest change' – consider quantifying

9) Line 286: 'small shift' – consider quantifying

10) Line 288: 'more moderate' – consider quantifying

11) Line 294: consider referring to Table 2 here

12) Figure 5: are these mean or median values?

13) Line 355: 'good correlation' – consider quantifying

14) Line 363: 'fairly similar', 'modest differences' – consider quantifying

15) Line 365, 368: 'fairly well', good agreement' – consider quantifying

16) Line 394: 'moved to insoluble' – do you mean moved to soluble?

17) Line 434: 'large particle concentrations' – do you mean number concentrations, what size range?

18) Line 468: a word seems to be missing before the words 'model produces'

---

## Referee Comment (RC2) · Anonymous Referee #2 · 26 Aug 2020

This manuscript presents a new physically-based parameterisation for in-cloud aerosol scavenging for use in size-resolved aerosol schemes within atmospheric models. The approach aims to model the scavenged fraction more explicitly than the common approach of using fixed coefficients for each size/composition/solubility class of aerosol. An evaluation of the scheme is presented in terms of the impact on vertical distributions of both number and mass, using measurements from the ATom flight campaign as a reference. These show that the scheme as initially applied leads to excess aerosol aloft with too long a lifetime, but a number of sensitivity studies show that re-tuning the size and solubility of aerosol at emission leads to much better performance.

The manuscript is clear and well presented, linking well to previous works on scavenging and the vertical distribution of aerosol, and is well suited to publication in GMD,

subject to the minor comments listed below:

**Detailed comments**

**Line 84:** is there any significance to liq/ice sometimes appearing as superscript and sometimes as subscript? I would suggest consistently using one or the other, or clearly explaining the difference in notation if this is significant.

**Lines 181–194:** please explain how SALSA fits into the framework of ECHAM-HAMMOZ, ECHAM and HAM, as this is not mentioned here and thus unclear.

**Line 229:** ECMWF does not make meteorological observations. Please clarify if this refers to a specific archive of third-party observations curated by ECMWF, or (as I suspect is more likely) to a *reanalysis* product such as ERA-5 or ERA-Interim rather than actual observations. Please cite the relevant dataset if possible.

**Line 289:** less ⟶ fewer.

**Lines 298–299 and 304:** please clarify that this specifically refers to ECHAM-HAMMOZ using SALSA (the widely-used modal scheme may behave differently).

**Figures 5 and 6:** please include baserun_old as a reference on these plots – otherwise it's hard to judge how the new scheme compares to the old against the actual observations.

**Table 2:** consider including a measure of the AEROCOM spread as well as its mean (or median) here; otherwise it's hard to say how "significantly" outside the pack a configuration is.

**Line 401:** some discussion of the caveats involved in assuming that the AEROCOM mean is the right target to tune towards would be welcome here.

---

## Author Comment (AC1) · 8 Oct 2020

We thank the Executive editor and both of the referees for raising to our attention important points. We are certain that after addressing these points in the revised manuscript have improved the quality of the manuscript. We have addressed all the points raised by the editor and referees and have marked the relevant corrections in blue in the revised manuscript.

**Answer to Executive editor short comment #1**

According to the Executive editor some technical requirements have not been met in our manuscript and thus we have revised those points. From here below we discuss the "Short comments" of the Executive editor. The editor's comments are marked in black and our answers are marked in blue.

Short comments:

- "The main paper must give the model name and version number (or other unique identifier) in the title."
- Code must be published on a persistent public archive with a unique identifier for the exact model version described in the paper or uploaded to the supplement, unless this is impossible for reasons beyond the control of authors. All papers must include a section, at the end of the paper, entitled "Code availability". Here, either instructions for obtaining the code, or the reasons why the code is not available should be clearly stated. It is preferred for the code to be uploaded as a supplement or to be made available at a data repository with an associated DOI (digital object identifier) for the exact model version described in the paper. Alternatively, for established models, there may be an existing means of accessing the code through a particular system. In this case, there must exist a means of permanently accessing the precise model version described in the paper. In some cases, authors may prefer to put models on their own website, or to act as a point of contact for obtaining the code. Given the impermanence of websites and email addresses, this is not encouraged, and authors should consider improving the availability with a more permanent arrangement. Making code available through personal websites or via email contact to the authors is not sufficient. After the paper is accepted the model archive should be updated to include a link to the GMD paper.

1. Thus add the model name and version number SALSA 2.0 in the title of your article.

   We have modified the title of the article as: "In-cloud scavenging scheme for sectional aerosol modules - Implementation in the framework of SALSA2.0 global aerosol module"

2. We very much appreciate that, while HAMMOZ is license restricted, a stand-alone version of SALSA 2.0 is made available. However, please note, that GMD is demanding authors to provide a persistent access to the exact version of the source code used for the model version presented in the paper. As explained in https://www.geoscientificmodel-development.net/about/manuscript_types.html the preferred reference to this release is through the use of a DOI which then can be cited in the paper. For projects in GitHub a DOI for a released code version can easily be created using Zenodo, see https://guides.github.com/activities/citable-code/ for details.

   We have added the DOI using Zenodo for the SALSA2.0 to the "Code availability" section as follows: "The stand-alone zero-dimensional version of SALSA2.0 is distributed under the Apache-2.0 licence and the code is available at https://github.com/UCLALES-SALSA/SALSA-standalone/releases/tag/2.0 (last access: 23 May 2018, Kokkola et al., 2018b) with DOI https://doi.org/10.5281/zenodo.1251668". We have also moved the ECHAM-HAMMOZ model revision text "The model data can be reproduced using the model revision r5511 from the repository https://redmine.hammoz.ethz.ch/projects/hammoz/repository/changes/echam6-hammoz/branches/fmi/fmi_trunk (last access: 8 March 2019, HAMMOZ consortium, 2019). The settings for the simulations are given in the same folder, in folder "gmd-2020-220"." from the "Data availability" section to the "Code availability" section.

3. Finally note, that according to our new Editorial (v1.2) all data and analysis / plotting scripts should be made available.

   We have added DOI for our data and analysis/plotting scripts to the "Data availability" section and modified the text as follows: "The data for reproducing the figures and codes for the figures can be obtained directly from authors or from https://etsin.fairdata.fi/dataset/f3cb5807-66fe-4a0d-a20a-ac208d3aab5a (last access: 29 June 2020, Holopainen et al., 2020) with DOI https://doi.org/10.23729/301df277-8147-4700-8652-ca491f2b58a".
   In addition, we have added DOI for the ATom aircraft measurements in the "Data availability" section as follows: "ATom aircraft data can be obtained through the Oak Ridge National Laboratory (ORNL) Distributed Active Archive Center (DAAC) https://daac.ornl.gov/cgi-bin/dsviewer.pl?ds_id=1581 (last access: 25 November 2019, Wofsy et al., 2018) with DOI https://doi.org/10.3334/ORNLDAAC/1581.".

**Answer to Referee #1**

According to the referee's comments the contents of the manuscript have been revised. Major points by the referee were that the aspects of model description and discussion lacked clarity and thus these aspects are revised. The new aspects of our wet deposition scheme are also revised as well as clearer presentation of the recommendations for the global modelling community.

From here below we discuss the "Specific Comments" and "Technical Corrections" of the referee. The referee's comments and corrections are marked in black and our answers are marked in blue.

Referee comments #1:

Specific Comments
1. An identification of the model used in the study would be of help to readers of the abstract.

   This is a good point and we have added the details of the model and its version to the abstract as follows: "We used the latest release version of ECHAM-HAMMOZ (ECHAM6.3-HAM2.3-MOZ1.0) with SALSA microphysics package to test and compare our scheme."

2. Line 10-11: Please clarify what sizes are meant by 'small particles'. As well, the number of these particles could be influenced by changes in the rate of new particle formation. As a result, it is not clear that this decrease in number concentration indicates that impaction scavenging is increased relative to the fixed coefficient scheme.

   In the revised manuscript, we have added a clarification to the sizes meant by 'small particles' as follows: "On the other hand, the number concentrations of particles smaller than 100 nm in diameter show a decrease, especially in the Arctic region.". The sensitivity studies in an article from Croft et al. (2010) state that impaction scavenging is the mechanism mostly affecting the number of smallest particles. This would indicate that the impaction scavenging is increased in the current scheme in comparison to the fixed coefficient scheme. However, the condensation sink reduces the new particle formation and thus also reduces the concentrations of particles smaller than 100 nm. Thus, we have modified the sentence starting from line 10 as follows: "These results could indicate that, compared to fixed scavenging coefficients, nucleation scavenging is less efficient, resulting in an increase of the number concentration of particles larger than 100 nm. In addition, changes in rates of impaction scavenging and new particle

formation (NPF) can be the main cause of reduction of the number concentrations of particles smaller than 100 nm.".

3. Lines 15-16: Why was the simulation baserun_old excluded from the comparisons with observations? Please consider including this simulation in comparisons with the observations.

   This is a good point and we have added the vertical profiles of baserun_old to figures 5 and 6 to the revised manuscript to illustrate the difference between the old and new schemes. In addition, we have modified the sentence at line 15-16 as follows: "Vertical profiles of aerosol species simulated with the scheme which uses fixed scavenging rates and the above mentioned sensitivity simulations were evaluated against vertical profiles from aircraft observations.".

4. Line 61: 'new in-cloud scheme' - Please consider clearly identifying the main aspects of the scheme that are new relative to previous studies. The word 'new' is used 5 times in this paragraph and repeatedly throughout the manuscript. It could be helpful to the readers to provide information that assists with understanding the developments made here relative to earlier work.

   Since we also use parts of the modal scheme of ECHAM-HAMMOZ, instead of "new", we will use terms "our" and "current", in the revised manuscript. We have elaborated the new aspects in the last paragraph of the Introduction. These aspects refer to the calculations of in-cloud nucleation scavenging by using the fraction of activated particles from the cloud activation scheme in liquid phase clouds (described in Section 2.1) and using the surface area of particles for calculations of removal fractions in the case of ice clouds (described in Section 2.2). These calculations of in-cloud nucleation scavenging are new in the framework of ECHAM-HAMMOZ for both modal and sectional aerosol microphysics modules. In the case of the sectional microphysics module, the new aspects also refer to the calculations of in-cloud impaction and below-cloud scavenging in a more size dependent method, as it was done in Croft et al. (2010) and Croft et al. (2009) (described in Section 2.3).

5. Eq. 1 – Are certain of these variables in-cloud versus grid-box mean?

   The values in Eq. 1 are in-cloud values and we will clarify this in the revised manuscript.

6. Line 107: Is the aerosol diameter wet or dry for this calculation?

   The diameters for this calculation are wet diameters and we have added the term "wet" to the text.

7. Line 113: What aerosols are ice nuclei in the model?

In the model, only particles which include mineral dust and black carbon are considered as ice nuclei and the way this is treated in ECHAM-HAMMOZ is described in detail by Hoose et al. (2008). We have added this to the end 3rd paragraph of Section 2.3 in the revised manuscript as follows: "In our model, only particles which include mineral dust and black carbon are considered as ice nuclei (Lohmann et al., 2007).".

8. Line 133-135: Is there a model version number? Please clarify what you mean by 'its sensitivity'.

The referee is correct here as more clarification is needed here. Thus, we have modified the text as follows: "To test how the in-cloud wet deposition scheme affects simulated global aerosol concentrations, we used it with the Sectional Aerosol module for Large Scale Applications version 2.0 (hereafter referred to as SALSA) in our ECHAM-HAMMOZ global model simulations. In addition, we tested how sensitive the simulated aerosol concentrations are to emission sizes, mixing, and aging, when this scheme is used.".
As mentioned above, the exact model version of the coupled ECHAM-HAMMOZ is now given in the Abstract.

9. Line 149: 'refine the entire scavenging scheme'. Is below-cloud scavenging also modified? Are both convective and stratiform wet removal modified? Please provide clarification about the wet removal treatment for the stratiform versus convective clouds. Are there differences between these two? How is the cloud fraction parameterized for each for the purposes of wet removal? Are there differences in the assumed updrafts for cloud droplet activation for stratiform and convective clouds?

The below-cloud scavenging was in fact modified in our simulations following Croft et al. (2009) to be more size dependent. However studies have found that below-cloud scavenging does not account for total aerosol mass deposition budgets nearly as much as in-cloud scavenging and thus we have neglected the analysis of this in our study. We have added a short description of the below-cloud scavenging to the end of Section 2.3 as follows: "For below-cloud scavenging, we used the Croft et al. (2009) method, in which we approximated each size class to be a log-normal mode. The size dependent collection efficiency for rain and snow uses an aerosol and collector drop size parameterization described in detail in Croft et al. (2009). Several studies have found that below-cloud scavenging of aerosols does not contribute to the mass deposition budgets as much as in-cloud scavenging does (Croft et al., 2009, 2010; Flossmann and Wobrock, 2010). Thus, we did not analyse below-cloud scavenging separately in our simulations.".
In our current wet removal scheme, only the stratiform cloud case is modified. For the convective case the model uses prescribed parameters presented in detail in Bergman et al. (2012). The cloud fraction for all of the simulations is parameterized according to

Tompkins et al. (2002). The updrafts are the same for both stratiform and convective cloud cases.

10. Line 154-155: What size is meant by 'large particles'? What size is meant by 'fairly small'? Did you conduct any test simulations for dust without the modified activation scheme but with the revised wet removal?

This was admittedly ambiguously phrased and we have rephrased this in the revised manuscript as: "This is because for larger than 1 $\mu$m insoluble particles with thin soluble coating (for instance mineral dust) the insoluble fraction is ignored in the cloud activation calculation and for those particles the activation is calculated as would be calculated for particles with dry size of the soluble part of the particles, thus making them less prone for activation.". We did in fact conduct a simulation for mineral dust without the modified activation scheme with revised wet removal and it massively decreased the number of activated dust particles in the largest insoluble size classes, but the compounds studied here (BC, OC, and SO4) were not noticeably affected.

11. Eq. 5-7: Please clarify if this is wet aerosol radius.

In Eq. 5-7 the radius refers to wet aerosol radius and we have added the term "wet" to the text for these equations.

12. Line 165: 'assume each size class is a lognormal mode' – for consistency is this same assumption also made for the nucleation scavenging? In that case, are separate scavenging coefficients calculated for mass versus number?

For the nucleation scavenging we do not make the assumption that each size class is a log-normal mode. Instead the cloud activation scheme calculates the fraction of activated particles in each size class separately and thus we do not need to make this assumption. Separate scavenging coefficients are also calculated for number concentration and mass concentration in the case of impaction.

13. Eq. 7 and Eq. 8: Are there specific references for the collision efficiencies, terminal velocity and ice crystal radius used here?

The collision efficiencies, terminal velocity and ice crystal radius in Eq. 7 and Eq. 8 follow the calculations summarized in Croft et al. (2009). Originally, the more detailed calculations for these values were presented in Slinn et al. 1984, Pruppacher and Klett 1998 and Seinfeld & Pandis 1998.

14. Section 2.4: Please clarify if the SALSA module is coupled to ECHAM6.3-HAM2.3-MOZ1.0 for all simulations.

This is a good suggestion by the referee and we have added clarification to the beginning of Section 2.3 as follows: "SALSA is the sectional aerosol module of ECHAM-HAMMOZ global climate model." and to the end of Section 2.4 as follows: "SALSA global aerosol module is coupled in the ECHAM-HAMMOZ global climate model for all of the simulations presented in this study.".

15. Section 2.3-2.5: Please consider adding a description about the treatment of aerosol aging in the baserun_old and baserun_new. Is there any exchange between the soluble and insoluble classes in the baserun simulations?

We have added clarification in the manuscript to the beginning of Section 2.5 as follows: "The treatment of aerosol aging is identical in baserun_old and baserun_new, i.e. there is no artificial transfer of insoluble particles to soluble size classes. However, aerosol mass can be transferred from the soluble to the insoluble population through coagulation.".

Also consider adding brief discussion about the treatment for OA emissions, sulfate emissions and chemistry. As well, do the simulated particles grow by aqueous phase sulfate production? What is the treatment for removal of gas-phase particle precursors?

For details of aerosol emissions and chemistry, we will direct the reader to Kokkola et al., 2018 which describes the implementation of SALSA to ECHAM-HAMMOZ together with its evaluation and we will add a sentence to the beginning of Section 2.3 as follows: "Details for calculations of aerosol emissions and chemistry in SALSA are presented in Kokkola et al. (2018a).". Particles do grow by aqueous phase sulfate production. In the removal of gas-phase particle precursors, we assume that their uptake by cloud droplets follows Henry's law (Bergman et al. 2012).

Consider mentioning here that the model does not include secondary organic aerosol.

We will mention this in the revised manuscript at the end of Section 2.4.

16. What is the treatment of below-cloud wet removal in these simulations? In the subsequent discussions, consider addressing how these parameterizations impact your conclusions. Likewise, what is the treatment of dry deposition and how does that affect your conclusions?

The treatment of below-cloud wet removal fractions in our simulations are obtained from a prescribed lookup table for which the calculations are presented in detail by Croft et al. (2009). We have also added a short description of this to the end of Section 2.3. As was mentioned in Point 9., we did not analyze the impact of below-cloud as studies have found that below-cloud scavenging does not account for total aerosol mass deposition budgets nearly as much as in-cloud scavenging. The treatment of dry deposition is

presented in detail by Bergman et al. (2012) and it has not been modified for our simulations as this study focuses mainly on wet deposition. Thus, a more detailed analysis of the effect of dry deposition is beyond this study.

17. Please consider referring to Fig. 1 at the start of Section 2.5 to help the reader to better follow the details presented.

    This is a good point by the referee and in the revised manuscript, we have changed the order of Fig. 1 and Table 1 and referred to Fig. 1 at the end of the first paragraph in Section 2.5.

18. Table 1: The color and line style for baserun_old and BC_soluble are very similar. Please consider revising.

    This is a good remark from the referee and in the revised manuscript we have modified the color and line style of simulation BC_soluble in Table 1 and Fig. 5-6 to be more recognizable from baserun_old simulation.

19. Line 213-214: Does the model also include biofuel emissions?

    Yes, we use ACCMIP emission data in which biofuel emissions are included.

20. Line 232-234: Are the baserun_old and baserun_new simulations not coupled to ECHAM-HAMMOZ but the sensitivity simulations are coupled and why? The text did not appear clear on the related description. Are the cloud microphysics parameterizations that are relevant for the wet removal different between the baserun simulations and the sensitivity simulations?

    All of our simulations are coupled in ECHAM-HAMMOZ global climate model using the SALSA2.0 global aerosol module. Thus, the cloud microphysics stay the same for all of the simulations. For clarification we have modified the end of Section 2.6 as follows: "The analysis is made between the old and the current wet deposition scheme using the ECHAM-HAMMOZ global aerosol-climate model with the SALSA aerosol module. In addition, the sensitivity of the current scheme to emission sizes, aging, and hygroscopicity of BC-containing aerosol, is tested using ECHAM-HAMMOZ with SALSA."

21. Section 2.7: What are the size ranges for the SP2 and HR-AMS? Do you extract mass concentrations from the model with consideration to similar size ranges?

    The size range for SP2 is 90-550 nm in diameter and for HR-AMS the size range is 35-1500 nm but in ATom data they use a cut off diameter of 1 micrometer. In our model

simulations, we use the full size range for BC and particles smaller than 1.7 micrometer in diameter for OC and SO4.

22. Line 245: What is the size range for the total number concentration?

In ATom aircraft measurements the size range for total number concentration is 2.7-4755 nm.

23. Line 268: 'model accumulated BC' – why does this occur, over how many years, would the model eventually reach a steady state?

To our knowledge the model accumulates BC as there are no efficient removal mechanisms in the upper parts of the atmosphere, especially with respect to the Arctic region. We simulated around 4 years of accumulation before the model reached a steady state.

24. Line 279: 'impaction scavenging is faster' – how can this be determined? If impaction scavenging is implicit in the prescribed coefficients scheme, then impaction rates cannot be directly compared between the two schemes - and as well other processes such as new particle formation can affect the number concentration. A similar question arises regarding the statement at line 273 since nucleation scavenging rates cannot be directly compared if both nucleation and impaction are implicitly represented by the fixed coefficients.

The referee is correct here that impaction rates can not be directly compared and we have noted that this was a bit too ambitious statement. Thus, in the revised manuscript we have modified the statement as follows: "In addition, the changes in rates of NPF and impaction scavenging in our current scheme result in an increased removal of small aerosol particles and thus reduce concentrations even more.". We have also modified the sentence in line 273 as follows: "This can be explained by changes in nucleation scavenging in the current scheme which reduces the wet removal of large particles and thus increases the number concentration of large particles."

25. Figure 2: Please consider using the same scale for the relative differences for all Panels.

This is a good point and we have modified the relative difference to be the same scale for all of the panels in Fig. 2 in the revised manuscript.

26. Figure 3: What is the size range for Ntot?

The size range for total modelled number concentration is the full size range of the model, i.e. 3 nm - 10 micrometer.

27. Figure 4: Do these plots include both stratiform and convective wet deposition?

The plots in Fig. 4 include both stratiform and convective wet deposition.

28. Line 294: Are there observation-based lifetimes available from previous studies for OA and sulfate, in addition to the lifetimes for BC from Lund et al., 2018?

Kristiansen et al. (2016) have studied observed and modelled aerosol lifetimes and they state that accumulation mode sulfate aerosol lifetime is estimated to be around 14.3 days and they also state that models generally underestimate sulfate lifetimes. For OA, we are not aware of any acceptable observation-based study of lifetimes.

29. Line 305: Do you mean the global mean BC 'lifetimes' are spurious? The previous section did not show vertical profiles.

The referee is correct that the spurious behavior was meant for the lifetimes. Thus, we have modified the first sentence in Section 3.2 as follows: "As reported in the previous section, ECHAM-HAMMOZ, using the SALSA aerosol module, with the current, more physical scheme, in its default setup, produced spuriously long lifetimes of all aerosol compounds, especially BC.".

30. Figure 5: Please consider if it would be instructive to include baserun_old in this Figure.

This is a good suggestion, and we have now added the baserun_old to Fig. 5 and Fig. 6 in the revised manuscript. In addition, we have added evaluation for the vertical profiles from baserun_old simulation to the text in Section 3.2.

31. Line 335: How sensitive are the results to the assumed supersaturation?

Since cloud activation is very much dependent on the supersaturation reached in activation, results are expected to be sensitive to them. However, the Abdul-Razzak & Ghan scheme calculates the supersaturation reached at cloud droplet activation and thus in our simulations, they are not assumed.

32. Line 344: What is meant by 'simplified sulfate chemistry'?

The sulfate chemistry scheme follows Feichter et al. (1996) which makes several simplifying assumptions. For example, gas phase OH mixing ratio is assumed to follow a cosine function with maximum at noon. In the revised manuscript we have added citation to Feichter et al. (1996) for a more detailed description to the reader.

33. Line 370: 'same aerosol size distribution' – please consider if this information should be in the methods – is this assumption used for all simulations?

In the revised manuscript we have added the information to the end of Section 2.4 as follows: "In addition, the model assumes the same aerosol emission size distribution per compound and emission sector throughout the whole world." and as mentioned earlier in point 15, details of emission size distributions are given by Kokkola et al. (2018). This assumption is valid for all simulations.

34. Lines 392: Some of this discussion was confusing – '3 times larger than in baserun_new' – where is baserun_new shown in Fig. 7?

The referee is correct that the Fig. 7 does not show baserun_new, but here we meant that if we compare these wet deposition mass fluxes to Fig. 4 which shows baserun_new we can see a 3 times larger BC fraction. For clarification, we have modified the sentence at line 383-385 in the revised manuscript as follows: "The number fluxes in the soluble population for the different sensitivity simulations show most change in the two smallest size classes, which increase by a factor of approx. 1.3 in the insol2sol simulation and approx. 1.1 for BC_large and BC_soluble when compared to baserun_new (shown in Fig. 4)." We also modified the text in lines 390-392 as follows: "While for BC_large and BC_soluble the BC mass fraction in the medium-sized insoluble particles disappears, in BC_small the BC fraction in the 50 to 100 nm insoluble particles is about 3 times larger than in baserun_new (shown in Fig. 4).".

35. Line 419: Please consider presenting what are the new developments made with this wet deposition scheme relative to previous studies. As well, consider putting the study in context of previous work by presenting how the main findings of this study compare to previous similar model developments.

As mentioned above in point 4, we decided that phrasing "new" is a bit misleading and in that we have pointed out the differences between our method and previous work.

36. Line 425: Perhaps the following could be clarified in the methods – for all simulations does SALSA run with an on-line coupling to a certain version of ECHAMHAM-MOZ model and are the outputs used for the wet removal from the cloud droplet activation scheme and ice nucleation scheme from the same ECHAM-HAM-MOZ?

The referee is correct that more clarification is needed here and all of the simulation runs were done using the SALSA aerosol module which was coupled to the ECHAM-HAMMOZ global climate model. In the revised manuscript, for clarification, we have modified the text starting from line 423 as follows: "We used the SALSA microphysics scheme coupled with the ECHAM-HAMMOZ global aerosol-chemistry-climate model to evaluate the differences between the old and current

wet deposition scheme. In addition, we used ECHAM-HAMMOZ with SALSA to test the sensitivity of the simulated aerosol concentrations to model assumptions of emission sizes, mixing, and aging when the current in-cloud wet deposition scheme was used.".

37. Line 428: What are the main adaptations needed for this wet removal scheme relative to the Croft et al. (2010) scheme?

The main difference is that here we use the fraction of activated particles for each size class from the cloud activation scheme and it is not dependent on whether the model uses modal or sectional approach. Another difference is that we use the surface area of particles in each size class for calculating the fraction of removal by ice clouds.

38. Line 437: Please clarify how you know that '..impaction scavenging in the new scheme was faster'. If impaction was implicit in the prescribed coefficients scheme and new particle formation also influences particle number concentrations as noted in the text – how can this statement be justified?

As mentioned above in points 2 and 24 that this was a bit too ambitious a statement and we have now modified this to refer to Croft et al. (2010) where they state that impaction scavenging affects the small particle sizes most. In the revised manuscript we have modified this sentence as follows: "In addition, the changes in impaction scavenging rates in the current scheme compared to the original setup can reduce the number concentration of particles smaller than 100 nm (Croft et al. (2010)).".

39. Line 460-462: The implementation of insoluble to soluble transfer is dismissed as being unsuitable. However, this aging process is commonly included with various parameterizations in global models. Are the authors able to clarify why the particular parameterization used in this study was chosen?

The referee is correct that this is a common way for treating aerosol aging. However, there is no physical basis to transferring particles from insoluble size classes to soluble ones after the insoluble particles have accumulated a certain amount of soluble material on them. To our knowledge, this transfer has not been justified in any publication. Instead, emitting BC to soluble size classes would account for mixing of BC and soluble material in emissions.

Are there certain aspects of the parameterization that could be improved with future work to enable a representation of aging from the insoluble to soluble classes?

Partitioning of semivolatile organic compounds between gas and particle phase could have a significant effect on aerosol vertical profiles and we will study this effect in the near future.

Why did the chosen parameterization for aging perform so poorly for dust in these simulations?

The method of moving insoluble to soluble bins causes aging to become too fast for dust. Thus, dust particles activate faster and are then removed too fast.

What are the emitted dust sizes?

The emissions of dust are calculated online in the model and the size and amount of emitted dust depends on the wind speed and it is described in more detail by Tegen et al. (2002) with modifications following Cheng et al. (2008) and Heinold et al. (2016).

40. Line 466: 'failed to reproduce global aerosol fields adequately: : :'. This statement is quite general. Please clarify. Does this statement refer to all aerosols – or it is specifically for BC? For example, at line 413, OC is excluded as a skill indicator.

The referee is correct that the current wet deposition scheme fails especially for BC, but when looking at the lifetimes in Table 2., we can see that also the lifetimes for other compounds are anomalous with respect to AEROCOM models. We have added clarification to the revised manuscript as follows: "To conclude, even though the current in-cloud wet deposition scheme is more physically sound than using fixed scavenging coefficients, it failed to reproduce global aerosol fields adequately in the default setup of the host model. This can be seen from the spuriously long lifetimes of all aerosol species.".
Following this, we have removed the sentence "Therefore we do not use the modelled OC lifetimes as skill indicator for the sensitivity studies here."

41. Please consider including the main recommendations for future model development based on the findings of this study.

In the revised manuscript we have added future developments based on this study to the end of Section 4 as follows: "In the future, the model development should include the study of effects of the gas-particle partitioning of semivolatile compounds which could have a significant impact on the modelled aerosol vertical profiles. In addition, the issue of the level of mixing of BC with soluble compounds during emissions and in the subgrid scale processing should be further investigated.".

Technical corrections:

1) Line 5: 'aerosol size' - please clarify if this is wet or dry aerosol radius

We have modified the text as follows: "For in-cloud impaction scavenging, we used a method where the removal rate depends on the wet aerosol size and cloud droplet radii."

2) Line 33: 'no or small amount' – consider removal of 'no'

We have modified the text as follows: "Transport of aerosol particles to remote regions with only small amounts of emitted particles, affects the local aerosol size distribution and composition (Rasch et al., 2000; Croft et al., 2010).".

3) Line 95-96: Check order of citations

We have changed the order of citations as follows: "(Stier et al., 2005; Seland et al.,2008; de Bruine et al., 2018)."

4) Line 48: Please check this citation as Ladino et al., 2011 appears to focus on impaction as opposed to nucleation scavenging.

We have corrected it as follows: "This process is called in-cloud nucleation scavenging (Pruppacher and Klett, 1997)."

5) Line 125: 'amount of nucleated ice particles' – do you mean number nucleated?

We have corrected this as follows: "Since we assume that the number of nucleated ice particles depends only on the aerosol surface area, the scavenging coefficient in ice-containing clouds in size class i is proportional to the ratio between nucleation rate in the size class and the total nucleation rate."

6) Figure 1: 'N' on vertical axis is not defined

We have added definition to the text as follows: "A schematic of the aerosol emission number size distribution, (N), as a function of diameter Dp, for the different simulations is presented in Fig. 1". In addition, we have added the definition to the caption of Fig. 1 as follows: "Schematic representation of the number size distribution, (N), of aerosols in different simulations as a function of diameter Dp.".

7) Line 259 'are lowest' – do you mean relative to other latitudes?

We have modified the text as follows: "In the tropics, these differences in the profiles are smaller, compared to the other latitude bands, with a maximum relative difference of approx.

200 % for BC and OC and slightly exceeding 150 % for SO4.". In addition, we have modified the text in line 263 as follows: "The Arctic shows the largest differences in the vertical profiles in comparison to the other latitude bands."

8) Line 283: 'modest change' – consider quantifying

We have added quantifying as follows: "There are only modest changes in the mass fluxes between the old and the current schemes. In the soluble population the largest difference is in the size class which spans diameters between 190-360 nm, where the current scheme exceeds the value of the old scheme by 0.003 µg/m2s. On the other hand, in the size class 1.7-4.1 µm, the old scheme has a higher value by 0.002 µg/m2s. In the insoluble population the current scheme exceeds the value of the old scheme by approx. 0.002 µg/m2s in the size class 190-360 nm, but in the largest size class the value of the old scheme is higher by 0.005 µg/m2s."

9) Line 286: 'small shift' – consider quantifying

We have rephrased this as follows: "In addition, there is a small increase of approx. 10^6 #/m2s in the current scheme in the size class between 190-360 nm.".

10) Line 288: 'more moderate' – consider quantifying

We have corrected this as follows: "For larger than 360 nm size classes the changes are insignificant."

11) Line 294: consider referring to Table 2 here

We have added a sentence to line 295 as follows: "The lifetimes for different compounds can be found in Table 2."

12) Figure 5: are these mean or median values?

The values presented here are mean values and we have added the word "mean" to all figure captions.

13) Line 355: 'good correlation' – consider quantifying

We have added quantifying as follows: "In the tropics, the simulations show a good correlation with the measurements as almost all of the profiles follow the shape of the profile of the ATom aircraft measurements, except for the surface concentrations, which are underestimated by a factor of approx. 2.5 compared to the measurements."

14) Line 363: 'fairly similar', 'modest differences' – consider quantifying

We have modified this to: "The Ntot profiles are similar in shape in all sensitivity simulations, with only a modest difference (600 #/cm3 at maximum), mostly at higher altitudes.".

15) Line 365, 368: 'fairly well', good agreement' – consider quantifying

We have added quantifying to these lines as follows: "In the mid-latitudes, all of the simulations represent Ntot concentrations fairly well (approximately 500 #/cm3 underestimation and 4000 \#/cm$^3$ overestimation at most) when compared to the measurements" and "At higher altitudes, starting from approx. 600 hPa upwards, insol2sol underestimates Ntot least, showing quite a good agreement with the measurements with only around 300 #/cm3 difference at most.

16) Line 394: 'moved to insoluble' – do you mean moved to soluble?

Yes, this is what we mean to say here and we have modified the text as follows: "In insol2sol, most of the BC is transferred from the insoluble to the soluble aerosol population before removal, which can be seen in a strong decrease in removed insoluble aerosol number for that simulation.".

17) Line 434: 'large particle concentrations' – do you mean number concentrations, what size range?

The referee is correct here and we have rephrased the sentence and the following sentence: "The current scheme also showed a significant increase of up to 600 % at maximum in the number concentration of particles larger than 100 nm which was similar in shape to the change in aerosol compound mass. However, the number concentration of particles smaller than 100 nm decreased everywhere, with a maximum decrease of 90 % in the Arctic.".

18) Line 468: a word seems to be missing before the words 'model produces'

The referee is correct here and for clarification we have modified the text as follows: "Based on the results of our sensitivity simulations, the ECHAM-HAMMOZ global climate model with SALSA aerosol module produces the best vertical profiles and aerosol lifetimes with the current scheme if BC is mixed with more soluble compounds at emission time.".

References

1. Bergman, T., Kerminen, V.-M., Korhonen, H., Lehtinen, K. J., Makkonen, R., Arola, A., Mielonen, T., Romakkaniemi, S., Kulmala, M.,and Kokkola, H.: Evaluation of the sectional aerosol microphysics module SALSA implementation in ECHAM5-HAM

aerosol-climate model, Geoscientific model development, 5, 845–868, https://doi.org/10.5194/gmd-5-845-2012, 2012.

2. Cheng, T., Peng, Y., Feichter, J., and Tegen, I.: An improvement on the dust emission scheme in the global aerosol-climate model ECHAM5-HAM, Atmospheric Chemistry and Physics, 8, 1105–1117, https://doi.org/10.5194/acp-8-1105-2008, https://acp.copernicus.org/articles/8/1105/2008/, 2008.

3. Croft, B., Lohmann, U., Martin, R. V., Stier, P., Wurzler, S., Feichter, J., Posselt, R., and Ferrachat, S.: Aerosol size-dependent below-cloud scavenging by rain and snow in the ECHAM5-HAM, Atmospheric Chemistry and Physics, 9, 4653–4675, https://doi.org/10.5194/acp-9-4653-2009, 2009.

4. Croft, B., Lohmann, U., Martin, R. V., Stier, P., Wurzler, S., Feichter, J., Hoose, C., Heikkilä, U., van Donkelaar, A., and Ferrachat, S.: Influences of in-cloud aerosol scavenging parameterizations on aerosol concentrations and wet deposition in ECHAM5-HAM, Atmospheric Chemistry and Physics, 10, 1511–1543, https://doi.org/10.5194/acp-10-1511-2010, 2010.

5. Flossmann, A. I. and Wobrock, W.: A review of our understanding of the aerosol–cloud interaction from the perspective of a bin resolved cloud scale modelling, Atmospheric Research, 97, 478 – 497, https://doi.org/https://doi.org/10.1016/j.atmosres.2010.05.008, 2010.

6. Heinold, B., Tegen, I., Schepanski, K., and Banks, J. R.: New developments in the representation of Saharan dust sources in the aerosol–climate model ECHAM6-HAM2, Geoscientific Model Development, 9, 765–777, https://doi.org/10.5194/gmd-9-765-2016, https://gmd.copernicus.org/articles/9/765/2016/, 2016.

7. Hoose, C., Lohmann, U., Bennartz, R., Croft, B., and Lesins, G.: Global simulations of aerosol processing in clouds, Atmospheric Chemistry and Physics, 8, 6939–6963, https://doi.org/10.5194/acp-8-6939-2008, 2008.

8. Kokkola, H., Kuhn, T., Laakso, A., Bergman, T., Lehtinen, K. E. J., Mielonen, T., Arola, A., Stadtler, S., Korhonen, H., Ferrachat, S.,Lohmann, U., Neubauer, D., Tegen, I., Siegenthaler-Le Drian, C., Schultz, M. G., Bey, I., Stier, P., Daskalakis, N., Heald, C. L., and Romakkaniemi, S.: SALSA2.0: The sectional aerosol module of the aerosol-chemistry-climate model ECHAM6.3.0-HAM2.3-MOZ1.0,Geoscientific Model Development, 11, 3833–3863, https://doi.org/10.5194/gmd-11-3833-2018, 2018.

9. Kristiansen, N. I., Stohl, A., Olivié, D. J. L., Croft, B., Søvde, O. A., Klein, H., Christoudias, T., Kunkel, D., Leadbetter, S. J., Lee, Y. H.,Zhang, K., Tsigaridis, K., Bergman, T., Evangeliou, N., Wang, H., Ma, P.-L., Easter, R. C., Rasch, P. J., Liu, X., Pitari, G., Di Genova, G., Zhao, S. Y., Balkanski, Y., Bauer, S. E., Faluvegi, G. S., Kokkola, H., Martin, R. V., Pierce, J. R., Schulz, M., Shindell, D., Tost, H., andZhang, H.: Evaluation of observed and modelled aerosol lifetimes using radioactive tracers of opportunity and an ensemble of 19 global models, Atmospheric Chemistry and Physics, 16, 3525–3561, https://doi.org/10.5194/acp-16-3525-2016, 2016.

10. Pruppacher, H. R. and Klett, J. D.: Microphysics of clouds and precipitation., Kluwer Academic Publishers, Dordrect, Boston, London, UK, 1998.

11. Seinfeld, J. H. and Pandis, S. N.: Atmospheric Chemistry and Physics, Wiley, New York, USA, 1998.

12. Slinn, W. G. N.: Precipitation Scavenging in Atmospheric Science and Power Production, CH. 11, edited by: Randerson, D., Tech. Inf. Cent., Off. of Sci. and Techn. Inf., Dep. of Energy, Washington DC, USA, 466–532, 1984.

13. Tegen, I., Harrison, S. P., Kohfeld, K., Prentice, I. C., and Coe, M., and Heimann, M.: Impact of vegetation and preferential source areas on global dust aerosol: Results from a model study, J. Geophys. Res., 107, 4576, doi:10.1029/2001JD000963, 2002.

14. Tompkins, A. M.: A Prognostic Parameterization for the Subgrid-Scale Variability of Water Vapor and Clouds in Large-Scale Models and Its Use to Diagnose Cloud Cover, Journal of the Atmospheric Sciences, 59, 1917–1942, https://doi.org/10.1175/1520-0469(2002)059<1917:APPFTS>2.0.CO;2, 2002.

According to the referee's comments the contents of the manuscript have been revised. From here below we discuss the "Detailed comments" of the referee. The referee's comments are marked in black and our answers are marked in blue.

Referee comments #2:

Detailed comments

1. Line 84: is there any significance to liq/ice sometimes appearing as superscript and sometimes as subscript? I would suggest consistently using one or the other, or clearly explaining the difference in notation if this is significant.

   The referee is correct here as more clarification is needed in the notations. Thus, for clarification, in the revised manuscript, we have corrected all of the liq/ice phrases for the equations, and the text, to be in subscript.

2. Lines 181–194: please explain how SALSA fits into the framework of ECHAM-HAMMOZ, ECHAM and HAM, as this is not mentioned here and thus unclear.

   ECHAM-HAMMOZ uses both modal (M7) and sectional (SALSA) microphysics representations of aerosol populations which can be selected before the model simulations. We have added a clarification to the beginning of Section 2.3 as follows: "SALSA is the sectional aerosol module of ECHAM-HAMMOZ global climate model." and to the end of Section 2.4 as follows: "SALSA global aerosol module is coupled in the ECHAM-HAMMOZ global climate model for all of the simulations presented in this study.".

3. Line 229: ECMWF does not make meteorological observations. Please clarify if this refers to a specific archive of third-party observations curated by ECMWF, or (as I suspect is more likely) to a reanalysis product such as ERA-5 or ERA-Interim rather than actual observations. Please cite the relevant dataset if possible.

   The referee is correct here and in the revised manuscript we have modified the text in line 229 as follows: "The model vorticity, divergence and surface pressure were nudged towards ERA-Interim reanalysis data provided by ECMWF (EuropeanCentre for Medium-Range Weather Forecasts) (Simmons et al., 1989; Berrisford et al., 2011), and the sea surface temperature (SST) and sea ice cover (SIC) were also prescribed."

4. Line 289: less −! Fewer.

   We have corrected this to be "fewer" instead of "less" in the revised manuscript.

5. Lines 298–299 and 304: please clarify that this specifically refers to ECHAM-HAMMOZ using SALSA (the widely-used modal scheme may behave differently).

   We have modified the lines 298-300 in the revised manuscript as follows: "Consequently, also the ability of ECHAM-HAMMOZ global climate model, with SALSA aerosol module, to reliably simulate aerosol vertical profiles and long range transport of aerosol is decreased when using the more physical scheme with the default model setup." for clarification. In addition, we have modified the lines 304-305 as follows: "As reported in the previous section, ECHAM-HAMMOZ, using the SALSA aerosol module, with the current, more physical scheme, in its default setup, produced spuriously long lifetimes of all aerosol compounds, especially BC.".

6. Figures 5 and 6: please include baserun_old as a reference on these plots – otherwise it's hard to judge how the new scheme compares to the old against the actual observations.

   This is a good remark from the referee and in the revised manuscript we have added baserun_old to Fig. 5 and Fig. 6 for more specific comparison with the measurements. In addition, we have added evaluation for the vertical profiles from baserun_old simulation to the text in Section 3.2.

7. Table 2: consider including a measure of the AEROCOM spread as well as its mean (or median) here; otherwise it's hard to say how "significantly" outside the pack a configuration is.

   This is a good point and in the revised manuscript we have added the spread of AEROCOM models to Table 2.

8. Line 401: some discussion of the caveats involved in assuming that the AEROCOM mean is the right target to tune towards would be welcome here.

   This is a good suggestion and in the revised manuscript we have added a sentence after the sentence starting at line 401 as follows: "However, we must keep in mind that AEROCOM means are global climate model based results and thus it is not completely certain that these lifetimes of different compounds reflect the actual lifetimes in the real atmosphere.".

---

## Author Comment (AC3) · 8 Oct 2020

Please find our answers in the supplement.

Please also note the supplement to this comment:
https://gmd.copernicus.org/preprints/gmd-2020-220/gmd-2020-220-AC3-supplement.pdf